# Functional dichotomy in spinal- vs prefrontal-projecting locus coeruleus modules splits descending noradrenergic analgesia from ascending aversion and anxiety in rats

Stefan Hirschberg[1], Yong Li[1], Andrew Randall[1,2], Eric J Kremer[3], Anthony E Pickering[1]*

[1]School of Physiology, Pharmacology and Neuroscience, University of Bristol, Bristol, United Kingdom; [2]Medical School, University of Exeter, Exeter, United Kingdom; [3]IGMM, CNRS, University of Montpellier, Montpellier, France

**Abstract** The locus coeruleus (LC) projects throughout the brain and spinal cord and is the major source of central noradrenaline. It remains unclear whether the LC acts functionally as a single global effector or as discrete modules. Specifically, while spinal-projections from LC neurons can exert analgesic actions, it is not known whether they can act independently of ascending LC projections. Using viral vectors taken up at axon terminals, we expressed chemogenetic actuators selectively in LC neurons with spinal ($LC^{:SC}$) or prefrontal cortex ($LC^{:PFC}$) projections. Activation of the $LC^{:SC}$ module produced robust, lateralised anti-nociception while activation of $LC^{:PFC}$ produced aversion. In a neuropathic pain model, $LC^{:SC}$ activation reduced hind-limb sensitisation and induced conditioned place preference. By contrast, activation of $LC^{:PFC}$ exacerbated spontaneous pain, produced aversion and increased anxiety-like behaviour. This independent, contrasting modulation of pain-related behaviours mediated by distinct noradrenergic neuronal populations provides evidence for a modular functional organisation of the LC.
DOI: https://doi.org/10.7554/eLife.29808.001

*For correspondence:
tony.pickering@bristol.ac.uk

**Competing interests:** The authors declare that no competing interests exist.

## Introduction

Neuropathic pain has a population prevalence of 7–8% (*Bouhassira et al., 2008*; *Torrance et al., 2006*) and remains difficult to treat (*Finnerup et al., 2015*) mandating research to improve therapies (*von Hehn et al., 2012*; *Skolnick and Volkow, 2016*). Many of the frontline treatments for neuropathic pain modulate spinal noradrenaline, either directly such as noradrenaline re-uptake inhibitors or as part of a common effector pathway such as gabapentinoids and opioids (*Nakajima et al., 2012*; *Hayashida et al., 2007, 2008*; *Jasmin et al., 2003*). A functional deficit of noradrenergic control is associated with the development of sensitisation in neuropathic pain models (*Hughes et al., 2013*; *Hughes et al., 2015*; *De Felice et al., 2011*; *Xu et al., 1999*). However, current systemic pharmacological interventions to correct this imbalance commonly produce CNS side-effects such as anxiety, sleep disturbance, mood shifts and confusion, limiting their therapeutic utility (*Finnerup et al., 2015*).

The locus coeruleus (LC) is the principal noradrenergic nucleus in the CNS (*Dahlstroem and Fuxe, 1965*) and is the main source of noradrenergic innervation to the spinal dorsal horn, forming part of a well-described analgesic circuit (*Millan, 2002*; *Pertovaara, 2006*; *Howorth et al., 2009a*; *Bruinstroop et al., 2012*). The LC also projects throughout almost the entire neuraxis and plays a

pivotal role in diverse behaviours such as learning and memory (*Takeuchi et al., 2016*; *Martins and Froemke, 2015*), strategic behaviour (*Tervo et al., 2014*), motivation (*Varazzani et al., 2015*), and arousal (*Sara and Bouret, 2012*; *Carter et al., 2010*; *Vazey and Aston-Jones, 2014*). Furthermore, LC-derived noradrenaline is also causally linked to aversive states like stress, anxiety and depression (*McCall et al., 2015*; *Alba-Delgado et al., 2013*). Thus, drug treatments for chronic pain that globally enhance noradrenaline levels or mimic its action are likely to cause adverse effects by acting on LC neuronal circuits that are not directly involved in pain control.

The LC has been characterised as a homogeneous cluster of neurons that provide a uniform global signal as part of a central arousal system (*Fuxe et al., 2010*). There is, however, accumulating evidence for specialisation of the LC neurons into subsets on the basis of: (1) anatomical projections (*Loughlin et al., 1986*; *Kebschull et al., 2016*; *Chandler et al., 2014*); (2) electrical properties (*Chandler et al., 2014*); and (3) co-transmitter content (*Simpson and Lin, 2007*). In accord with this organisational principle, we have demonstrated that distinct populations of LC noradrenergic neurons innervate the spinal cord (**LC:SC**) and prefrontal cortex (**LC:PFC**) based on anatomy and electrophysiology *in vitro* and *in vivo* (*Li et al., 2016*). Here, we selectively target LC neurons from their projections to the spinal cord or prefrontal cortex using the axon terminal uptake and retrograde transport of canine adenovirus type 2 (CAV-2) vectors (*Junyent and Kremer, 2015*; *Li et al., 2016*). This CAV-2 vector expresses a chemogenetic actuator in noradrenergic neurons specifically, to allow activation of **LC:SC** or **LC:PFC** neurons to test whether the beneficial noradrenergic analgesia and stress-like adverse effects seen in pain states are mediated by distinct subpopulations of LC neurons and are therefore dissociable.

## Results

### Targeting and activation of noradrenergic LC neurons with PSAM

To enable selective activation of LC neurons, the excitatory chemogenetic receptor PSAM (*Pharmaco- Selective Actuator Module, PSAM$^{L141F,Y115F:5HT3HC}$* [*Magnus et al., 2011*]) was expressed in noradrenergic neurons (*Figure 1A–B*) using viral vectors with the cell-type-specific synthetic promotor PRSx8 (PRS) (*Hwang et al., 2001*; *Lonergan et al., 2005*; *Howorth et al., 2009a*). Efficient and selective targeting of LC neurons was observed after direct injection into the dorsal pons of a lentiviral vector harbouring the PRS-EGFP-2A-PSAM expression cassette. After transduction, EGFP immunofluorescence was detected in 690 ± 38 neurons per LC (N = 5) of which 98 ± 0.4% co-expressed dopamine β-hydroxylase (DBH) confirming their identity as noradrenergic (*Figure 1B*). To enable post-hoc immuno-localisation, PSAM was tagged with a C-terminal HA. Subsequently, PSAM-HA was detected in the membranes of the LC neurons (*Figure 1C*, 82/112 transduced neurons examined from three rats). The functionality of the PSAM-HA was verified by Ca$^{2+}$-imaging in cell lines (*Figure 1—figure supplement 1*).

Patch clamp recordings from transduced LC neurons in acute pontine slices showed that the designer-ligand PSEM308 (*pharmaco-selective effector molecule*) produced inward currents and a reversible excitation. PSEM308 evoked concentration-dependent inward currents of up to 800 pA in voltage clamp (V$_{hold}$60 mV, 1–10 μM, *Figure 1D*). Current-clamp recordings of transduced LC neurons showed an eight-fold increase in firing frequency (0.54 ± 0.21 Hz to 4.22 ± 0.90 Hz, *Figure 1E*) in response to PSEM308 (3 μM). No response to PSEM308 (doses up to 10 μM) was detected in recordings from non-transduced LC neurons (n = 6). We found no difference in intrinsic electrical properties between transduced and non-transduced LC neurons (*Figure 1—figure supplement 2*) suggesting that expression of PSAM is well tolerated. Together, these findings indicate that PSAM is functionally expressed in LC neurons and its activation is sufficient to robustly increase action potential discharge.

### Chemogenetic excitation of the LC in vivo

Extracellular recordings with multi-barrelled electrodes were used to determine whether PSEM308 application could excite LC neurons in vivo (after direct LC transduction with LV$^{PRS-EGFP-2A-PSAM}$ (*Figure 1F*)). In PSAM expressing rats (N = 8), local pressure application of PSEM308, but not saline, caused a time-locked and reproducible excitation in 12 out of the 24 recorded units (*Figure 1F*), increasing their firing to 234% of baseline (2.63 ± 0.92 Hz to 6.15 ± 1.47 Hz). The remaining cells did

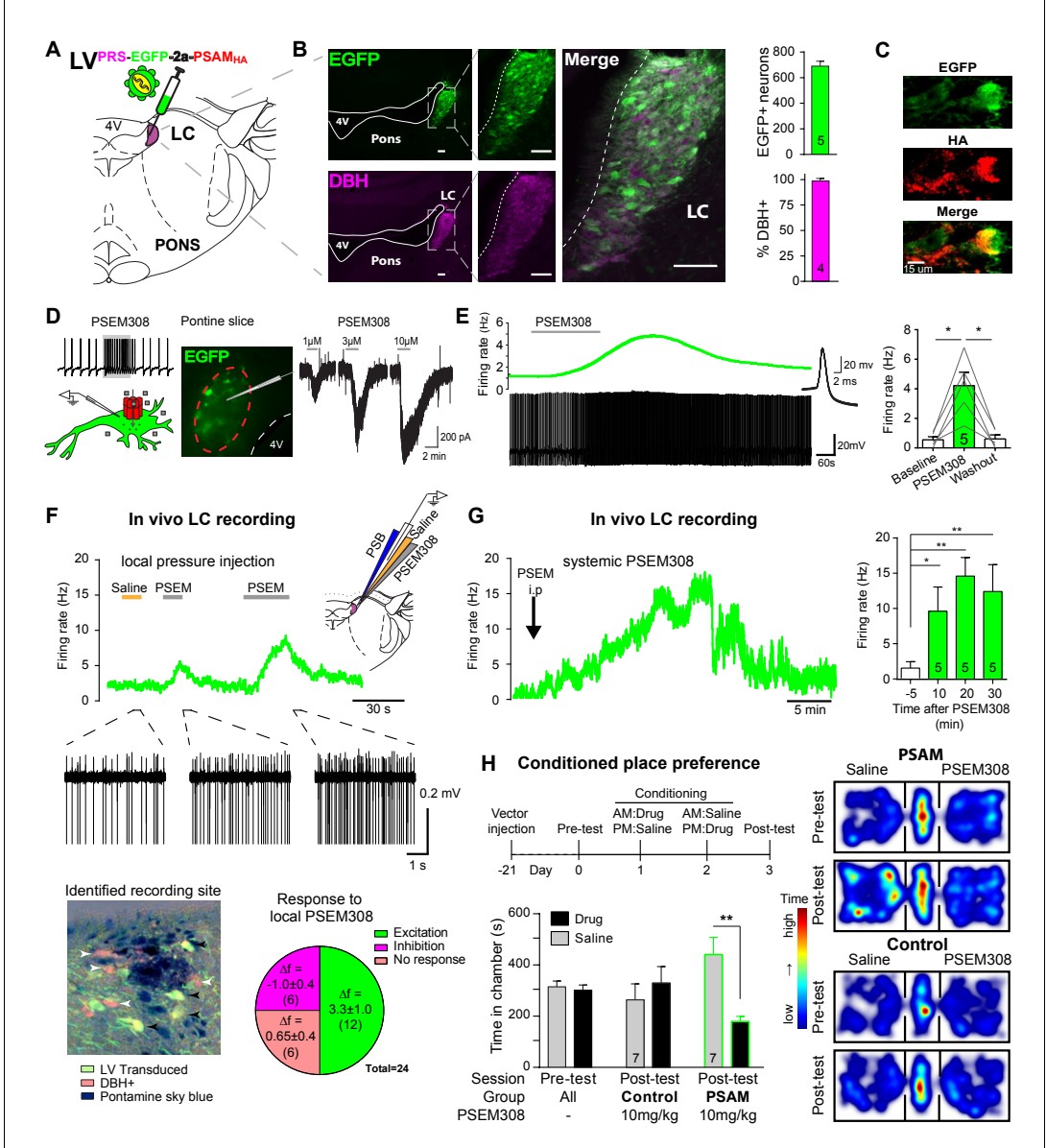

**Figure 1.** PSAM-mediated chemogenetic activation of LC neurons in vivo. (**A**) Strategy using direct stereotaxic injection of lentiviral vector to express the excitatory ionophore PSAM in noradrenergic LC neurons. (**B**) Selective transduction of LC demonstrated by immunohistochemistry (IHC) for EGFP and dopamine β-hydroxylase (DBH) with 690 EGFP+ neurons per LC of which 98% were DBH+ (scale bar 100 μm). (**C**) PSAM expression was demonstrated using IHC for the HA tag. (scale bar 15 μm) (**D**) Schematic of PSEM308-mediated excitation of transduced neurons expressing PSAM. Patch clamp recordings from EGFP+ LC neurons in acute pontine brain slices. Perfusion of PSEM308 evoked concentration-dependent inward currents. (**E**) PSEM308 (3 μM) increased rate of firing of transduced LC neurons. Inset shows 16 overlaid action potentials. Group data shows increase in firing produced by PSEM308 (3 μM). (**F**) Extracellular recordings from LC neurons in anaesthetised rats using multi-barrel recording electrodes allowing local pressure ejection of PSEM308/saline/pontamine sky blue (PSB). Traces show graded excitation of an identified LC neuron by PSEM308. Recording sites were subsequently histologically identified by the PSB staining within the LC (transduced cells identified by IHC for EGFP (black arrowheads) and DBH (white arrowheads). The response to local PSEM308 was categorised as excitation or inhibition if it changed firing rate by more than 3 SD from the baseline rate. Application of PSEM308 produced an excitation in 50% of LC neurons. We found a second group of neurons that showed no response, presumably as they were not transduced. A third group showed an inhibition of spontaneous firing in response to local PSEM308 application. (**G**) Kinetics of the excitatory response to systemic PSEM308 administration (10 mg/kg i.p). (**H**) Timeline of conditioned place aversion protocol to assess influence of chemogenetic activation of LC neurons on behaviour. In PSAM expressing rats, PSEM308 (10 mg/kg) caused conditioned place aversion but had no effect on control animals. Representative heat maps of rat position in the pre-test and post-test after PSEM308 with bilateral LC transduction with LV^PRS-EGFP-2A-PSAM HA or control LV^PRS-EGFP All data analysed with repeated measures ANOVA (one or two way as appropriate) with Bonferroni's post hoc testing (*p<0.05, **p<0.01). (See also *Figure 1—figure supplements 1*, *2* and *3*)

*Figure 1 continued on next page*

*Figure 1 continued*

DOI: https://doi.org/10.7554/eLife.29808.002

The following source data and figure supplements are available for figure 1:

**Source data 1.** Figure 1 source data.
DOI: https://doi.org/10.7554/eLife.29808.006
**Figure supplement 1.** Generation of a traceable version of the PSAM in an expression cassette with enhanced fluorescence.
DOI: https://doi.org/10.7554/eLife.29808.003
**Figure supplement 2.** Transduction with PSAM does not alter LC neuronal electrophysiological properties.
DOI: https://doi.org/10.7554/eLife.29808.004
**Figure supplement 3.** Simultaneous recording of one excited and one inhibited LC neuron in vivo with focal PSEM308 (1 mM) pressure application.
DOI: https://doi.org/10.7554/eLife.29808.005

not respond (n = 6) or showed a small reduction of their firing frequency (−34.6%, 2.83 ± 1.21 to 1.85 ± 0.86 Hz, n = 6) in response to local PSEM308 (*Figure 1F*). These inhibited units, which were in some cases simultaneously recorded along with LC units whose activity was increased (*Figure 1—figure supplement 3*), likely reflect LC neurons that are not themselves transduced but are inhibited by noradrenaline released from surrounding transduced and hence PSEM308-activated LC neurons (as previously hypothesised by *Vazey and Aston-Jones, 2014*). These data are in line with the observed distribution of cells at the recording sites, which were always surrounded by DBH expressing neurons of which many, but not all, were also EGFP+ (*Figure 1F*). Local application of PSEM308 did not alter baseline firing frequency in control LC neurons (2.99 ± 1.55 to 3.15 ± 1.58 Hz, NS, n = 7 neurons from three non-transduced rats).

The effect of systemic application of PSEM308 was tested on transduced LC units that showed an excitatory response to local PSEM308 (n = 5, *Figure 1G*). Intraperitoneal (i.p) injection of PSEM308 (10 mg/kg, same dose used throughout the rest of the study unless otherwise stated) increased discharge frequency 10-fold from 1.47 ± 0.93 to 14.76 ± 2.8 Hz after 20 min. LC neurons doubled their firing frequency after 5.9 ± 3.9 min and cells reached their maximum discharge frequency after 15.9 ± 6.5 min (*Figure 1G*, median 15.8 Hz, range 10.5 to 27.2 Hz). The evoked excitation outlasted the time of experimental recording (30 min) in four out of five neurons.

## Chemogenetic activation of the LC alters behaviour

High tonic LC discharge frequencies (5–10 Hz) are acutely aversive (*McCall et al., 2015*). Therefore, we employed a Pavlovian-conditioned place preference (CPP) paradigm to assay whether there was a behavioural consequence to chemogenetic activation of LC neurons. CPP is a voluntary choice task using the innate behaviour of animals to divide their time between environments according to their affective valence. Conditioning with PSEM308 induced avoidance of the drug-paired environment in animals that bilaterally expressed PSAM in LC neurons (*Figure 1H*). Such avoidance was not seen with a lower dose of PSEM308 (5 mg/kg), or in animals transduced with a control vector (LV$^{PRS-EGFP}$) and given PSEM308 (10 mg/kg).

## Activation of LC:SC but not LC$^{:PFC}$ is anti-nociceptive

Based on previous studies, we predicted that **LC$^{:SC}$** neurons would play a role in anti-nociception (*Jones and Gebhart, 1986*; *Howorth et al., 2009b*) and hypothesised that the forebrain projecting LC neurons might play a different – potentially pro-nociceptive role (*Hickey et al., 2014*). To target noradrenergic neurons according to their projection territory, we inserted a PRS-EGFP-2A-PSAM expression cassette into a CAV-2 vector to target LC neurons (*Figure 2A*) from their axonal projections (*Li et al., 2016*; *Soudais et al., 2001*).

Bilateral injection of CAV2$^{PRS-EGFP-2A-PSAM}$ into the dorsal horn of the lumbar spinal cord transduced 380 ± 37 neurons in each LC (N = 7, *Figure 2A*). These **LC$^{:SC}$** neurons were located in the ventral aspect of the nucleus. In a second set of animals, CAV2$^{PRS-EGFP-2A-PSAM}$ bilaterally injected into the PFC transduced similar numbers of neurons (361 ± 72 per LC, N = 6). However, these **LC$^{:PFC}$** neurons were scattered throughout the LC with a more dorsal location than the **LC$^{:SC}$** neurons (representing different groups of neurons as previously noted [*Bruinstroop et al., 2012*; *Howorth et al., 2009a*; *Li et al., 2016*]). In both cases, the transduced **LC$^{:SC}$** and **LC$^{:PFC}$** neurons

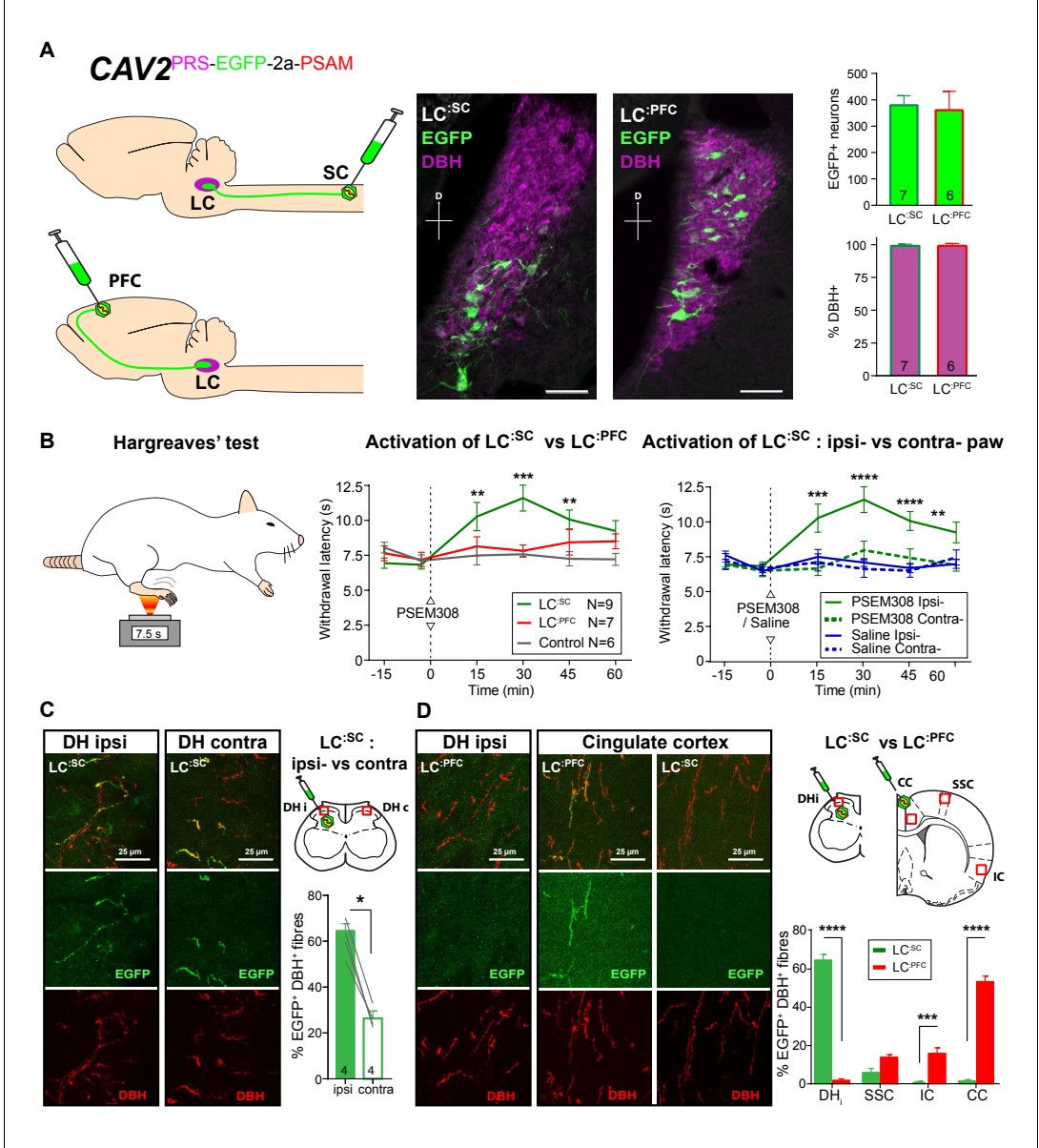

**Figure 2.** Activation of descending LC:SC but not ascending LC:PFC is anti-nociceptive. (A) Retrograde transduction strategy with canine adenoviral vectors (CAV2) to target noradrenergic LC neurons with projections to the spinal cord (LC:SC) or prefrontal cortex (LC:PFC). Similar numbers of LC neurons were transduced by injections in lumbar spinal cord (L3-4, 380) and prefrontal cortex (361). In both cases, >99% of neurons were DBH+. (scale bar 100 μm). (B) The Hargreaves' test (radiant heat) was used to measure hind- paw withdrawal latency. PSEM308 activation of LC:SC but not LC:PFC caused a robust anti-nociception (increase in withdrawal latency). The LC:SC anti-nociceptive effect was only seen in the ipsilateral hind paw (same side as spinal CAV2 injection). (C) Representative images of double immunofluorescence showing DBH-positive fibres in the spinal cord that were anterogradely filled with EGFP after transduction of LC:SC neurons. Quantification of the percentage of DBH+ fibres that were EGFP+ showed a three-fold higher density of EGFP-labelled fibres ipsilateral to vector injection (Mann-Whitney test, *p<0.05) which is consistent with the lateralized analgesic effect seen in (B). (D) Representative images and comparison of the percentage of EGFP labelled fibres in cortical areas and the spinal cord after LC:SC or LC:PFC transduction (N = 3 in each group). Data analysed with two-way repeated measures ANOVA with Bonferroni's post hoc tests (**p<0.01, ***p<0.001, ****p<0.0001).

DOI: https://doi.org/10.7554/eLife.29808.007

The following source data is available for figure 2:

**Source data 1.** Figure 2 source data.

DOI: https://doi.org/10.7554/eLife.29808.008

were positive for DBH (99.1 ± 0.5% and 99.1 ± 0.7%, respectively) indicating efficient and specific retrograde transduction of the noradrenergic neurons (*Figure 2A*).

We used the Hargreaves' test to measure hind paw thermal withdrawal latency as an assay of nociception. Chemogenetic activation of **LC:SC** neurons (PSEM308 i.p N = 7 rats) significantly increased thermal withdrawal latency to radiant heat as compared to naive rats (N = 6) (15 min post i.p., 10.3 ± 1.0 s **LC:SC** vs 7.5 ± 0.7 s naive, 30 min post i.p., 11.6 ± 0.9 s **LC:SC** vs 7.6 ± 0.2 s naive, 45 min post i.p., 10.1 ± 0.7 s **LC:SC** vs 7.3 ± 0.4 s naive [*Figure 2B*]). This time course of the anti-nociceptive effect, between 15 and 45 min post-dosing, is consistent with the kinetics of LC neuron activation after i.p PSEM308 injection (*Figure 1G*). Moreover, this **LC:SC** anti-nociception was only seen in the hind-limb ipsilateral to the vector injection in the spinal cord (*Figure 2B*), indicating an unexpected lateralised organisation of the **LC:SC** projection. By contrast, activation of **LC:PFC** neurons had no effect on the thermal withdrawal latency. This indicates a functional dichotomy in the actions of the ascending and descending LC projections on acute pain.

By taking advantage of the anterograde axonal EGFP fills of the retrogradely targeted LC neurones it was possible to examine the projection pattern of **LC:SC** and compare it to **LC:PFC**. This showed that although ipsilateral spinal injection of CAV2^PRS-EGFP-2A-PSAM labelled LC neurones on both sides, those neurones exhibited a lateralised projection to the spinal cord with a higher density of labelling seen on the side of the original CAV2 injection (Ipsilateral 62 ± 3.6% of DBH+ fibres also EGFP+ Vs contralateral 26 ± 2.3%, N = 4, *Figure 2C*). This is in line with the lateralised anti-nociception noted after activation of the **LC:SC** (*Figure 2B*). In contrast, the **LC:PFC** showed very sparse projection of axons to the spinal cord but a high density was seen in cortical sites which was greatest in the PFC (53 ± 2.9% EGFP+/DBH+).

To investigate the mechanism of this **LC:SC** chemogenetic anti-nociception, extracellular recordings were made from wide dynamic range (WDR) neurons in the spinal dorsal horn (n = 14 from six rats, mean depth 415 μm, range 200 to 600 μm) in a preparation that allows the application of drugs directly onto the spinal cord in the irrigating solution (*Figure 3A*), (*Funai et al., 2014*; *Furue et al., 1999*). WDR neurons encode stimulus intensity, responding to both innocuous and noxious stimuli (*Price and Dubner, 1977*). The average spontaneous discharge frequency of WDR was 1.86 ± 0.60 Hz, that was increased to 16.52 ± 2.19 Hz by pinch stimulation of the hind paw. In **LC:SC** transduced animals, perfusion of PSEM308 (10 μM) inhibited the pinch-evoked discharge two-fold to 8.5 ± 1.8 Hz (12/14 WDR neurons, *Figure 3D*). However, both spontaneous firing and innocuous, brush-evoked discharge were unaffected by PSEM308 (*Figure 3D*). These data demonstrate that local activation of terminals of **LC:SC** neurons in the spinal cord causes a specific inhibition of nociceptive inputs to WDR neurons.

To address the question of whether the **LC:SC** inhibition of the pinch response was caused by spinal release of noradrenaline, Atipamezole (10 μM, α2-adrenoceptor antagonist) was co-applied with PSEM308 (10 μM). Atipamezole prevented the **LC:SC** mediated-inhibition of the pinch response (*Figure 3D*). Atipamezole alone had no effect on spontaneous discharge, evoked discharge after pinch or brush indicating there is little ongoing spinal release of noradrenaline under anaesthesia (paired t-test p>0.05, n = 5 WDR neurons from two preparations, *Figure 3E*).

## Activation of LC:PFC module is aversive and anxiogenic

Because activation of **LC:PFC** neurons did not change nociceptive thresholds, we investigated whether activation of these neurons contributes to aversion or to the anxiety-like behaviour ascribed to the LC by *McCall et al. (2015)*. We found that activation of **LC:PFC** caused avoidance of the drug-paired chamber in the CPP paradigm indicating a negative affective valence (*Figure 4A*).

The open-field test uses the innate behaviour of rodents to avoid exposed areas as a measure of anxiety. Anxiogenic stressors typically reduce the time rodents spend in the centre of the arena and increase the time animals are immobile. Chemogenetic activation of **LC:PFC** reduced centre time by 62.6% and doubled immobility time in the open-field test after PSEM308 i.p (*Figure 4B*). These findings indicate that chemogenetic activation of the **LC:PFC** neurons changes behaviour to produce anxiety/aversion and promoted immobility (as previously reported for high-frequency stimulation of the whole LC [*Carter et al., 2010*]).

In contrast, when the **LC:SC** neurons were transduced, conditioning with PSEM308 was not aversive in the CPP paradigm (*Figure 4A*) and did not produce anxiety-like behaviour in the open-field test (*Figure 4B*). There was no difference in the total distance animals travelled between drug or

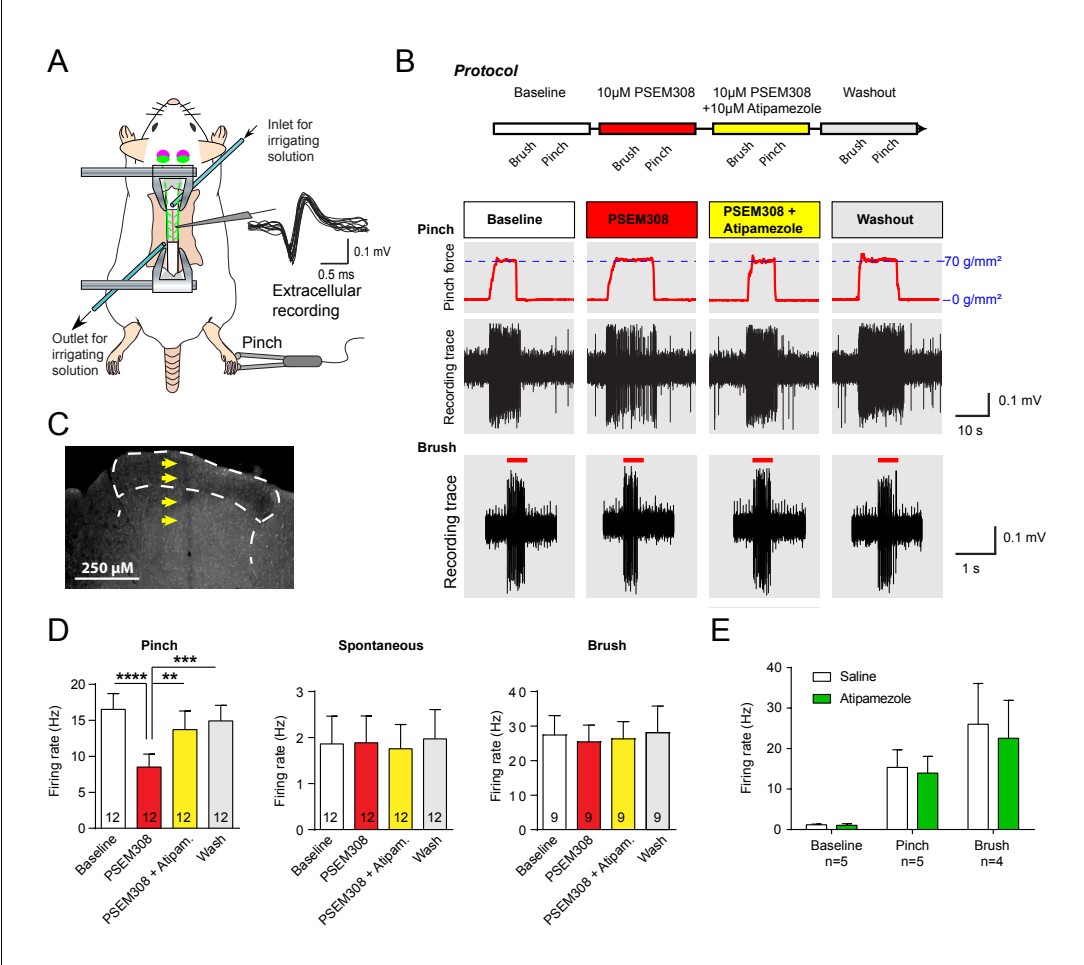

**Figure 3.** Acivation of LC$^{:SC}$ selectively supresses nociceptive excitation of wide dynamic range neurons of the dorsal horn via an alpha2-adrenoceptor in naive rats. (A) Spinal extracellular recording from wide dynamic range neurons in the dorsal horn. (B) Experimental protocol for noxious pinch (calibrated forceps) or non-noxious brush applied to the hindpaw during spinal drug application. Original traces of pinch or brush responses from WDR neurons during application of PSEM308 (10 µM) or PSEM308 (10 µM) + atipamezole (10 µM) (C) Posthoc histology of the spinal dorsal horn outlining the electrode track. (D) In LC$^{:SC}$ transduced rats, spinal superfusion of PSEM308 (10 µM) caused a significant attenuation of the response to noxious pinch which was prevented by Atipamezole (10 µM) indicating a spinal α2-adrenoceptor mechanism (two-way repeated measures ANOVA with Bonferroni's post hoc test, **$p<0.01$,. ***$p<0.001$, ****$p<0.0001$). Application of PSEM308 or atipamezole does not change spontaneous firing or brush evoked discharge of WDR neurons. (E) Quantification of neuronal responses to pinch, brush and spontaneous discharge frequency during saline or atipamezole (10 µM) application (paired t-test $p>0.05$). Data are presented as mean ±SEM.

DOI: https://doi.org/10.7554/eLife.29808.009

The following source data is available for figure 3:

**Source data 1.** Figure 3 source data.
DOI: https://doi.org/10.7554/eLife.29808.010

saline injection in the open-field test for LC$^{:SC}$ animals (1894±90 PSEM308 vs 1885 ± 137 cm saline, paired t-test p=0.94) suggesting there was no gross locomotor deficit. This indicates that chemogenetic activation of the LC$^{:SC}$ module does not produce the same negative affective behaviour that is seen with the LC$^{:PFC}$ module.

## Engagement of LC$^{:SC}$ module attenuates neuropathic sensitisation via α2-adrenoceptors

To test for possible analgesic benefit of chemogenetic activation of the LC$^{:SC}$ module in neuropathic pain, we employed the tibial nerve transection model (*Figure 5A* and *Figure 5—figure supplement 1*). This procedure causes the progressive development of punctate mechanical (von Frey threshold

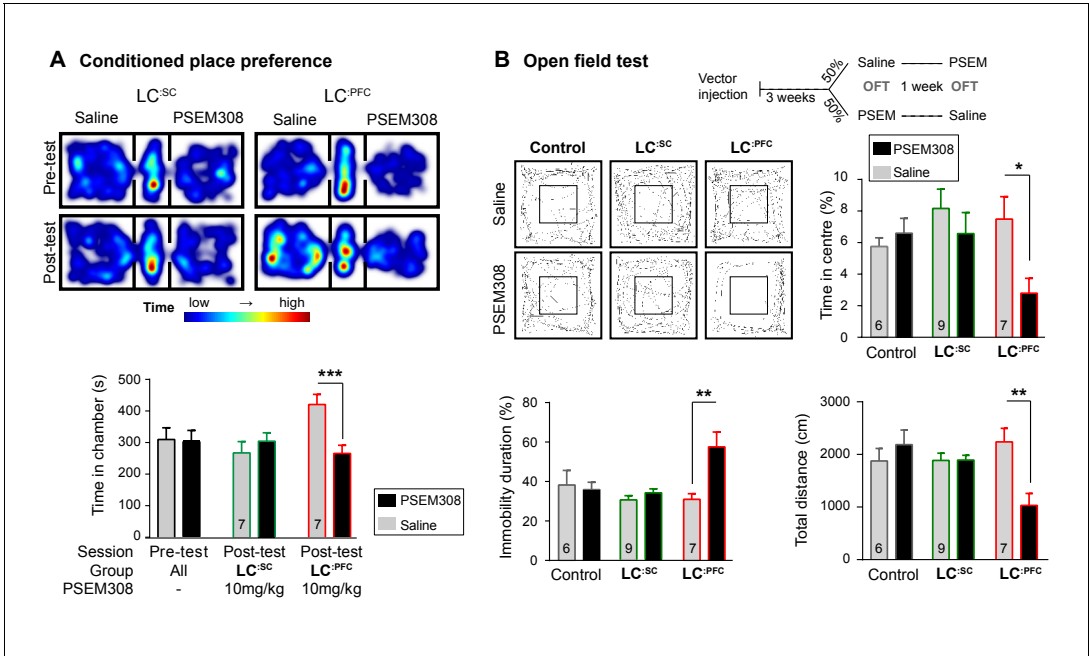

**Figure 4.** Chemogenetic activation of LC:PFC neurons is aversive and anxiogenic unlike LC:SC. (**A**) Representative heat maps showing rat location in the place preference arena before and after conditioning for LC:SC (left) and LC:PFC (right) transduced rats (outer compartments 30 × 30 cm). Activation of LC:PFC with PSEM308 (10 mg/kg) caused conditioned aversion to the paired chamber (two-way repeated measures ANOVA with Bonferroni's multiple comparison between the conditioned and unconditioned chamber, ***$p < 0.001$) and had no effect on LC:SC transduced rats. Pre-test data pooled for graphical presentation (**B**) Cross-over design for open-field test (OFT) and tracks showing activity after PSEM308 (10 mg/kg) or saline (inner square 30 × 30 cm). PSEM308 reduced the time LC:PFC rats spent in the centre of the arena and increased the time these rats were immobile. In contrast, PSEM308 had no effect on LC:SC transduced rats (two-way RM- ANOVA with Bonferroni's multiple comparison test, PSEM308 vs saline, *$p < 0.05$, **$p < 0.01$).

DOI: https://doi.org/10.7554/eLife.29808.011

The following source data is available for figure 4:

**Source data 1.** Figure 4 source data.
DOI: https://doi.org/10.7554/eLife.29808.012

lowered from 16.0 ± 1.5 to 1.3 ± 0.3 g, N = 15) and cold hypersensitivity (increased frequency of paw withdrawals from 10.6 ± 3.8 to 82.7 ± 3.8%, N = 15) over 3 weeks (*Figure 5—figure supplement 1*). Chemogenetic activation of the LC:SC module increased the mechanical withdrawal threshold after nerve injury (from 1.2 ± 0.3 baseline to 11.7 ± 3.3 g, 30 min after PSEM308 i.p, N = 6 (*Figure 5B*)). Similarly, we found that activation of LC:SC produced a significant improvement in weight-bearing on the nerve-injured limb, linking the reduced evoked mechanical hypersensitivity to a spontaneous behaviour (*Figure 5—figure supplement 1*). Rotarod testing of TNT animals showed that activation of the LC:SC had no effect on motor performance indicating that these actions were not a consequence of motor impairment and reflected an analgesic action (*Figure 5E*). The chemogenetic analgesic effect on mechanical and cold allodynia was dose-dependent (1–10 mg/kg) and completely blocked by prior intrathecal (i.t) injection of the *a*2-adrenoceptor antagonist yohimbine (60 ng, 10.0 ± 2.6 g PSEM308 + vehicle to 1.2 ± 0.4 g PSEM308 + yohimbine, N = 5 (*Figure 5D*)). In marked contrast to LC:SC, activation of the LC:PFC module by PSEM308 had no effect on evoked mechanical or cold responses in the nerve injury model and did not improve weight-bearing (*Figure 5B* and *Figure 5—figure supplement 1*).

These findings indicate that activation of LC:SC neurons is analgesic in this model of neuropathic pain and that the effect is mediated by noradrenaline acting on *a*2-adrenoceptors in the spinal cord. Therefore, we predicted that the noradrenaline reuptake inhibitor reboxetine (1 mg/kg i.p), at a dose too low to cause an analgesic effect on its own, would enhance chemogenetically evoked LC:SC analgesia. Co-administration of reboxetine amplified and prolonged the analgesic effect of PSEM308 and completely reversed mechanical hypersensitivity (17.8 ± 3.2 g) as compared to sham animals (17.9 ± 1.1 g, [*Figure 5C*]).

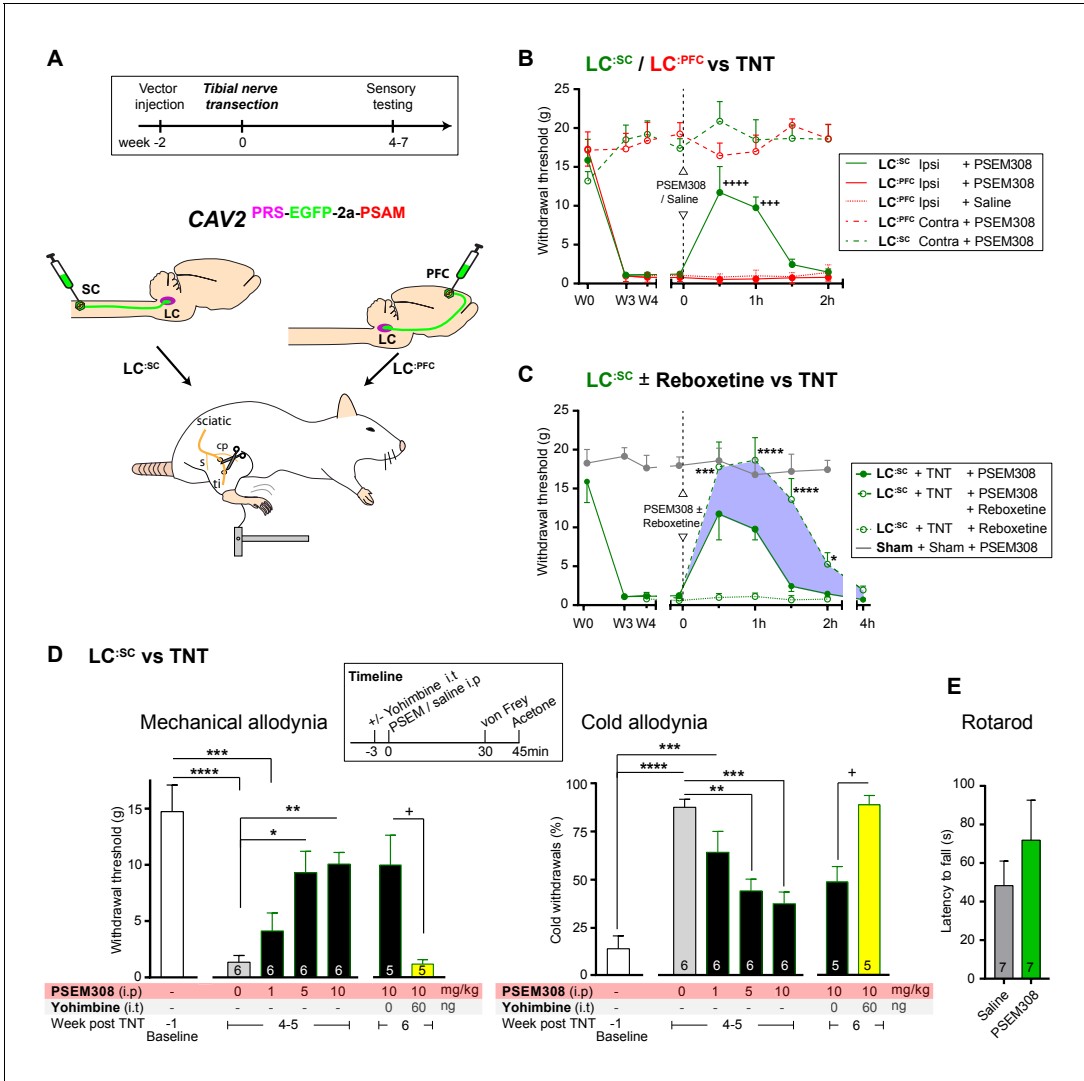

**Figure 5.** Chemogenetic activation of LC:SC is analgesic via a spinal a2-mechanism (behaviour). (A) Experimental timeline of vector injection and the tibial nerve transection model of neuropathic pain (s = sural, cp = common peroneal and ti = tibial nerves). (B–C) Quantification of mechanical withdrawal threshold (von Frey, after *Chaplan et al., 1994*). (B) Nerve injured animals showed a mechanical sensitisation of the ipsilateral limb. Activation of LC:SC neurons (PSEM308 (10 mg/kg)) attenuated the neuropathic sensitisation (comparison vs baseline +++p<0.001, ++++p<0.0001, N = 6 all groups). In contrast, activation of the LC:PFC had no effect on neuropathic sensitisation at any time point (C). Co-application of reboxetine i.t (1 mg/kg) enhanced the analgesic effect of LC:SC activation shown in panel (B). (D) Timeline of sensory testing in weeks 4–6. The analgesic effect of LC:SC activation on mechanical and cold allodynia is dose-dependent and is completely blocked by intrathecal yohimbine the α2-adrenoceptor antagonist (paired t-test, +p<0.05). (E) Activation of LC:SC had no effect on the latency to fall in the rotarod test - a measure of motor impairment (paired t-test, p>0.05). All analysis by repeated measures ANOVA (one- or two-way) with Bonferroni's multiple comparison, *p<0.05, **p<0.01, ***p<0.001, ****p<0.0001 unless otherwise stated (see also *Figure 5—figure supplement 1*)

DOI: https://doi.org/10.7554/eLife.29808.013

The following source data and figure supplement are available for figure 5:

**Source data 1.** Figure 5 source data.
DOI: https://doi.org/10.7554/eLife.29808.015

**Figure supplement 1.** Development of neuropathic pain phenotype after tibial nerve transection and analgesic profile of chemogenetic activation of LC:SC vs LC:PFC.
DOI: https://doi.org/10.7554/eLife.29808.014

To characterise the mechanism of this **LC:SC** chemogenetic analgesia in the TNT model, extracellular recordings were made from dorsal horn neurons in the L4 segment that showed a WDR profile (n = 13 neurons from five anaesthetised TNT rats) (*Figure 6*). These neurons responded to innocuous (von Frey 2 g and brush) and noxious (70 g/mm$^2$ pinch) mechanical stimuli as well as to acetone

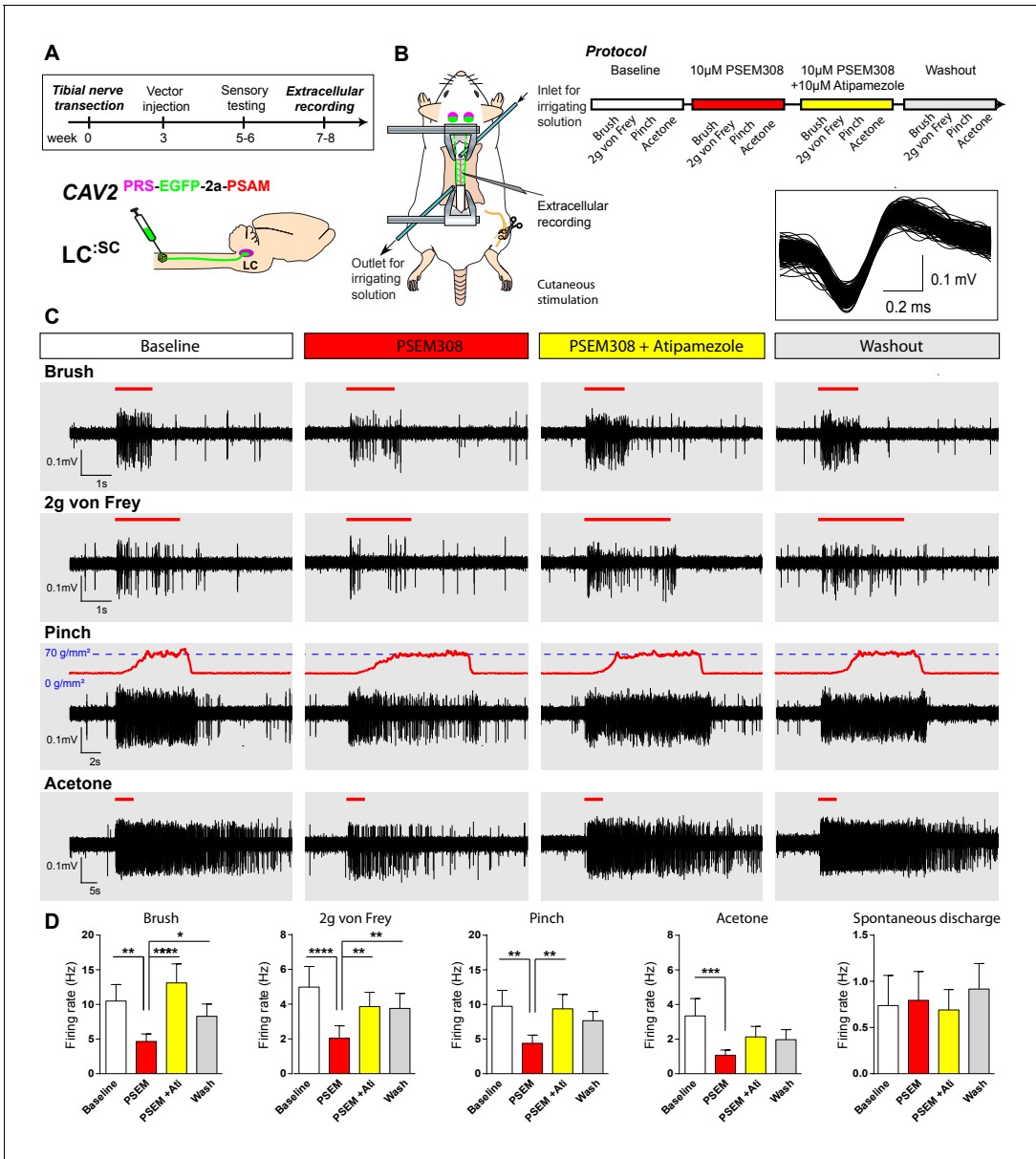

**Figure 6.** Chemogenetic activation of LC:SC is analgesic via a spinal α2-mechanism (cell recording). (**A**) Experimental timeline of vector injection and the tibial nerve transection. CAV2[PRS-EGFP-2A-PSAM] was injected to the spinal cord after the neuropathic phenotype was fully developed in week 3. (**B**) Spinal extracellular recording and experimental protocol. The insert shows the waveform of 500 action potentials from one discriminated unit from the recording traces below. (**C**) Representative recording traces (containing two discriminated WDR units) from the spinal dorsal horn with responses to mechanical and cold stimuli during spinal drug application. (**D**) Spinal superfusion of PSEM308 (10 μM) caused a significant attenuation of the response to all stimuli which was prevented by Atipamezole (20 μM) indicating a spinal a2-adrenoceptor mechanism. All data analysed with two-way repeated measures ANOVA with Bonferroni's post hoc testing (**p<0.01, ***p<0.001, ****p<0.0001).

DOI: https://doi.org/10.7554/eLife.29808.016

The following source data is available for figure 6:

**Source data 1.** Figure 6 source data.

DOI: https://doi.org/10.7554/eLife.29808.017

drops (cooling) applied to the hind paw. In LC:SC transduced animals, the response of WDR units to all stimuli was reversibly attenuated by spinal application of PSEM308 (10 µM) and this effect was blocked by concurrent perfusion of atipamezole (10 µM) (*Figure 6C+D*). In control, non-transduced, TNT rats PSEM308 was without effect on evoked responses in WDR neurons (n = 8) (*Figure 6— source data 1*). The spontaneous firing of the WDR neurons was unaffected by PSEM308 (*Figure 6D*). These data demonstrate that local activation of terminals of LC:SC neurons in the spinal cord of nerve injured animals causes a α2-adrenoceptor-mediated inhibition of both responses to innocuous and noxious stimuli in WDR neurons (unlike in naïve animals where only responses to noxious stimuli were attenuated) but has no effect on spontaneous activity. This is consistent with an inhibition of the afferent drives rather than a change in the excitability of the WDR neurons themselves.

## Bidirectional modulation of spontaneous pain and pain-related behaviours by LC modules

We assayed the frequency of spontaneous foot-lifts as an index of ongoing pain caused by nerve injury. As anticipated, we found that nerve injury significantly increased the number of spontaneous foot-lifts over 5 min (6.2 ± 0.7 sham, N = 6 vs 30.7 ± 3.9 TNT, N = 17 [*Figure 7B + Video 1*]). Activation of LC:SC more than halved the number of spontaneous foot-lifts in nerve injured rats (13.1 ± 2.9), consistent with a reduction of ongoing pain. By contrast, activation of LC:PFC neurons almost doubled the number of foot-lifts (52.6 ± 9.3 [*Figure 7B*]) suggesting a diametric modulation of spontaneous pain by these distinct subsets of LC neurons.

If this dichotomous action of the LC on spontaneous pain is also reflected in the affective experience of pain, then we anticipated that chemogenetic stimulation of the LC:SC module would reduce the negative pain-associated affect whilst LC:PFC activation should increase pain affect. In rats with neuropathic sensitisation, activation of LC:SC increased the time spent in the PSEM308-paired chamber, whereas LC:PFC activation induced aversion (*Figure 7A*). These data indicate that the neuropathic animals obtain benefit from LC:SC activation but that their situation is worsened by LC:PFC activation.

A link between the signs of ongoing pain and the affective component of pain is provided by the correlation between the PSEM308-induced modulation of chamber preference and the incidence of spontaneous foot-lifts across all groups of animals (*Figure 7C*). No correlation was found between chamber preference and PSEM308-induced changes in evoked responses (data not shown) suggesting that modulation of ongoing pain is a better predictor of the perceived benefit of this intervention.

The negative affect induced by LC:PFC activation was also evident from measures of anxiety in the open-field test. Activation of LC:PFC reduced the time animals spent in the centre of the arena by 63% (5.4 ± 1.6% saline to 2.0 ± 0.7% PSEM308, N = 7) and increased immobility by 139% compared to saline (47.3 ± 6.0% saline to 65.9 ± 7.7% PSEM308, N = 7 (*Figure 7—figure supplement 1*). Furthermore, the effect on immobility caused by activation LC:PFC neurons was further enhanced by 135% by co-administration of reboxetine (1 mg/kg) (61.3 ± 8.0% PSEM308 to 82.8 ± 3.9% PSEM308 + reboxetine, N = 6) indicating that the effect was caused by release of noradrenaline. In contrast, PSEM308 application had no effect on open-field performance in LC:SC transduced or sham animals (*Figure 7—figure supplement 1*).

## Transduction of the LC:SC after nerve injury still produces chemogenetic analgesia

In order to be a viable therapeutic approach for clinical neuropathic pain the retrograde transduction strategy to target the LC:SC would need to be effective after nerve injury. We have previously demonstrated that there is a relative depletion of DBH + fibres in the lumbar spinal segments after TNT which conceivably may impair vector transduction (*Hughes et al., 2013*). Therefore, as a proof-of-concept experiment we explored the feasibility of using CAV2PRS-EGFP-2A-PSAM transduction 3 weeks after nerve injury once sensitisation was maximal in the TNT model. Activation of the LC:SC neurones by systemic administration of PSEM308 (10 mg/kg) increased mechanical withdrawal thresholds, attenuated cold allodynia and greatly reduced the incidence of spontaneous paw flinches (*Figure 8B–D*) much as we observed with pre-emptive transduction. The change in evoked

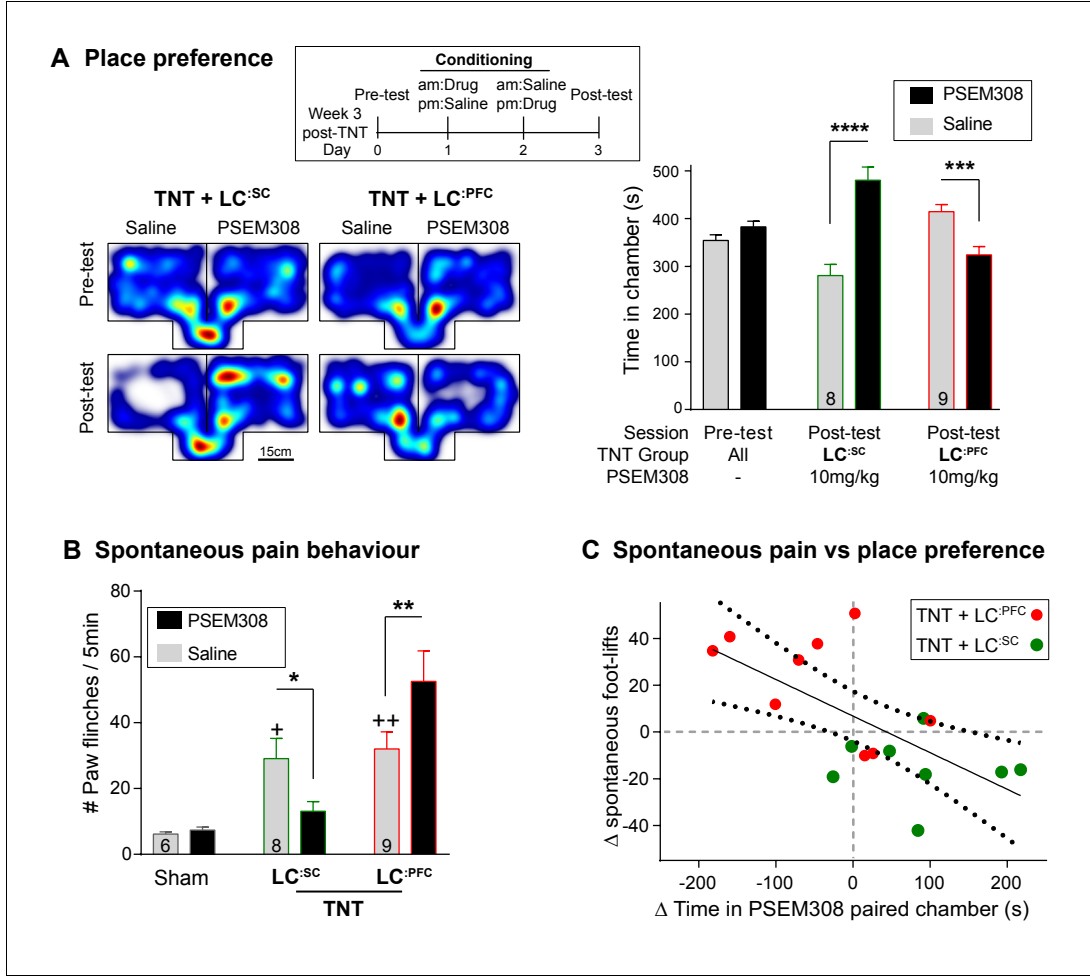

**Figure 7.** Chemogenetic activation of LC[:SC] is rewarding in neuropathic pain model whilst activation of LC[:PFC] neurons exacerbates spontaneous pain behaviour. (**A**) Timeline and representative location heat maps for a rat in CPP arena. In TNT animals, activation of LC[:SC] now induces a robust place preference, whereas activation of LC[:PFC] continues to produce aversion (two-way RM-ANOVA with Bonferroni's multiple comparison between the conditioned and unconditioned chamber, ***p<0.001, ****p<0.0001). (**B**) TNT animals showed an increased frequency of paw flinches - a measure of spontaneous pain (saline treatment, RM-ANOVA with Bonferroni's multiple comparison vs sham, +p<0.05, ++p<0.01). Activation of LC[:SC] reduced and LC[:PFC] increased the frequency of flinches (two-way RM-ANOVA with Bonferroni's multiple comparison PSEM308 (10 mg/kg) vs saline, *p<0.05, **p<0.01). (**C**) Across both LC[:SC] and LC[:PFC] groups there was a correlation between the degree of modulation of spontaneous pain and the chamber preference (r = −0.65, p<0.01). (See also *Video 1*).

DOI: https://doi.org/10.7554/eLife.29808.018

The following source data and figure supplement are available for figure 7:

**Source data 1.** Figure 7 source data.
DOI: https://doi.org/10.7554/eLife.29808.020

**Figure supplement 1.** Chemogenetic stimulation of the LC[:PFC] module is anxiogenic in chronic pain.
DOI: https://doi.org/10.7554/eLife.29808.019

responses was blocked by intrathecal atipamezole (50 µg) indicating that it was mediated by spinal activation of $a$2- adrenoceptors (*Figure 8C*). Post-hoc histology for native EGFP revealed an equivalent level of transduction of LC neurones following the spinal CAV2 administration (CAV2 before TNT - 164.8 ± 38.1, N = 8 vs CAV2 after TNT - 165.4 ± 25.5 neurons per animal, N = 7).

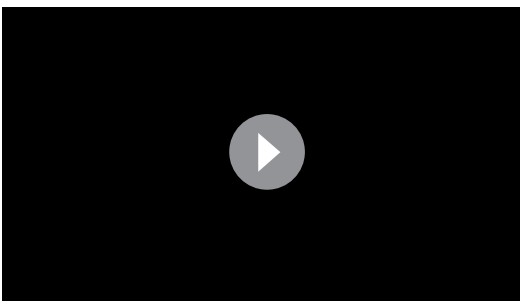

**Video 1.** Chemogenetic activation of LC:SC module suppresses spontaneous pain behaviour in neuropathic pain model.
DOI: https://doi.org/10.7554/eLife.29808.021

## Discussion

By modifying LC cells using an intersectional vector strategy, based on transcriptional profile and projection territory, we rendered defined subgroups of noradrenergic neurons chemogenetically-activatable both in vitro and in vivo. Selective activation of the **LC:SC** or **LC:PFC** modules produced distinct changes in behaviour that were functionally antithetical in the context of pain states. The activation of the **LC:SC** neurons evokes an *a*2-adrenoceptor-mediated anti-nociceptive action that is analgesic in a model of chronic neuropathic pain. This amelioration of the pain phenotype by **LC:SC** activation is seen for both evoked and spontaneous behaviour and was accompanied by reduced negative pain affect. In contrast, activation of the **LC:PFC**

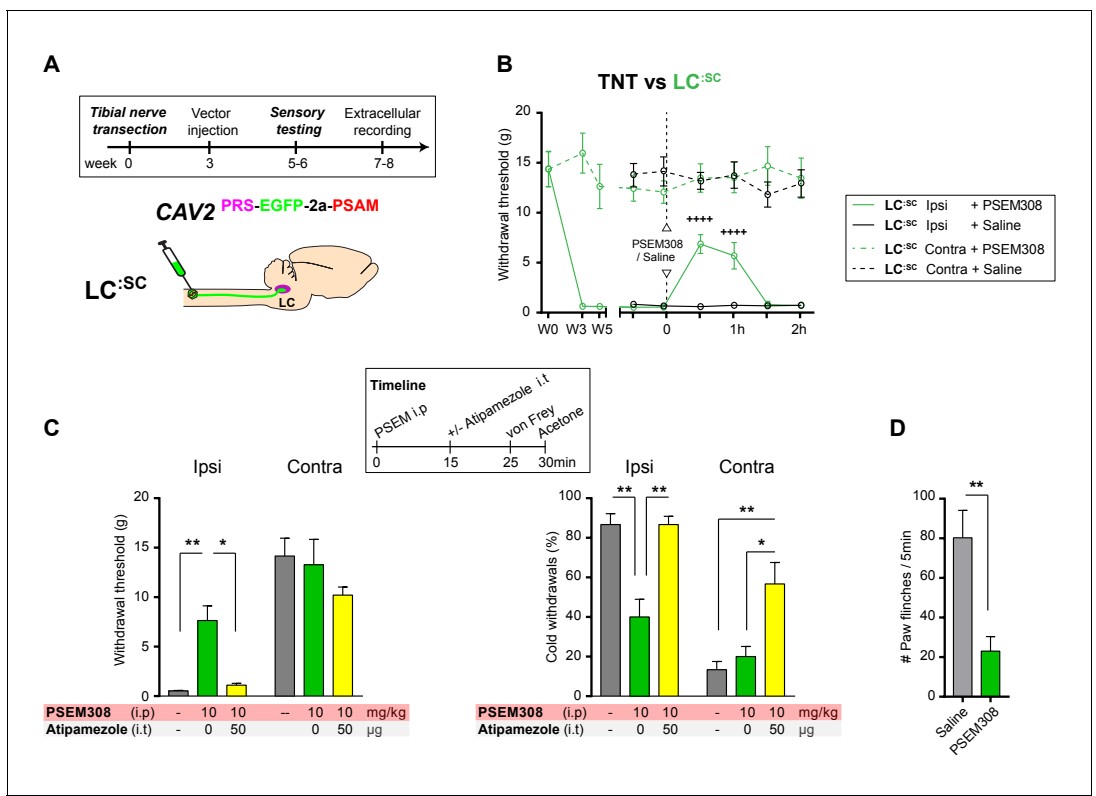

**Figure 8.** Chemogenetic activation of LC:SC as an intervention for established neuropathic pain. (**A**) Experimental timeline of vector injection and the tibial nerve transection. CAV2[PRS-EGFP-2A-PSAM] was injected after the neuropathic phenotype was fully developed in week 3. (**B**) Quantification of mechanical withdrawal threshold (von Frey, after *Chaplan et al., 1994*). Activation of **LC:SC** neurons (PSEM308 (10 mg/kg)) in week 5 attenuated the neuropathic sensitisation (repeated measures two-way ANOVA with Bonferroni's multiple comparison, ++++$p<0.0001$, N = 7). (**C**) Timeline of sensory testing in week 6. The analgesic effect of **LC:SC** activation on mechanical and cold allodynia is completely blocked by intrathecal atipamezole the *a*2-adrenoceptor antagonist (repeated measures two-way ANOVA with Bonferroni's multiple comparison, *$p<0.05$, **$p<0.01$, N = 7). (**D**) Activation of **LC:SC** reduced the frequency of spontaneous flinches (paired t-test, **$p<0.01$).
DOI: https://doi.org/10.7554/eLife.29808.022

The following source data is available for figure 8:

**Source data 1.** Figure 8 source data.
DOI: https://doi.org/10.7554/eLife.29808.023

produces an anxiety-like pattern of behaviour with negative affective valency and does not alter nociceptive sensitivity but instead increases expression of the signs of spontaneous pain after nerve injury.

Within the pain field, there has been a long interest in descending noradrenergic control which has been well characterised pharmacologically in vitro and in vivo (*Millan, 2002*; *Yoshimura and Furue, 2006*; *Pertovaara, 2006*; *Jones, 1991*). However, it has been difficult to selectively engage this system either clinically or in animal models because of its anatomical location, nuclear organisation, and lack of pharmacological targets to enable differentiated control with drugs. Here, we use a chemogenetic engineering approach (*Magnus et al., 2011*) to manipulate and investigate the system with systemic drug administration. In targeting the **LC:SC** neurons, we see an anatomical organisation that is similar to that previously described (*Bruinstroop et al., 2012*; *Howorth et al., 2009a*; *Li et al., 2016*; *Westlund et al., 1983*) with labelling in a cluster of around 15% of the LC neurons located in the ventral pole of the nucleus. **LC:SC** activation in vivo was anti-nociceptive in a heat withdrawal assay that was lateralised to the side of CAV2$^{PRS-EGFP-2A-PSAM}$ vector injection in the spinal cord. This lateralised anti-nociceptive effect was in line with a clear ipsilateral spinal predominance (three-fold) in the anterogradely filled noradrenergic fibres. Given that LC neurons were transduced bilaterally in the pons after this unilateral injection (as previously noted [*Howorth et al., 2009a*]), this indicates that there is a novel lateralised functional organisation for the control of nociception.

In a neuropathic pain model, activation of the **LC:SC** attenuated the degree of allodynia and hyperalgesia. This was seen for all sensory modalities tested and also applied to spontaneous measures of sensitisation i.e. foot-lifts. This beneficial effect of engaging the **LC:SC** extends the findings from previous chronic intrathecal dosing studies with noradrenaline re-uptake inhibitors that also suppressed the neuropathic phenotype (*Hughes et al., 2015*). Given that we had documented a depletion of LC fibres in the spinal cord at the level of the nerve afferent input (*Hughes et al., 2013*) then it was not known whether there would be the potential to restore noradrenergic inhibition with the retrograde chemogenetic strategy. The demonstration of preserved actions indicates that both release of noradrenaline and the *a*2-adrenoceptor mechanism required are still functional. Furthermore, by transducing the **LC:SC** neurons with CAV2 after nerve injury we could still produce potent chemogenetic suppression of sensitisation. This provides proof of concept that such a retrograde targeting approach may be viable as a therapeutic strategy after neuropathic pain has developed.

Pain is an emotive experience and relief of pain is a strong learning signal with positive affective valence whereby animals display a preference for conditions where their pain is alleviated (*King et al., 2009*). Therefore, the preference of nerve-injured animals for **LC:SC** activation indicates that this was analgesic rather than simply suppressing a reflex motor withdrawal – this parallels previous studies with intrathecal clonidine/noradrenaline re-uptake inhibitors that have shown a conditioned place preference (*Hughes et al., 2015*; *Wei et al., 2013*) or drive for self-administration (*Martin et al., 2006*). Additionally, no preference for **LC:SC** activation was seen in naive rats, indicating that it is the analgesic action in the neuropathic model that has a positive valence rather than intrinsic rewarding effect of **LC:SC** stimulation. This is consistent with the sparse forebrain innervation (particularly of reward-related sites) for these **LC:SC** neurons (*Howorth et al., 2009a*; *Li et al., 2016*).

Engagement of the forebrain projecting **LC:PFC** produced a contrasting profile without any action on evoked-nociception and evidence of a facilitation of spontaneous pain behaviours in the neuropathic pain model. Additionally, activation of the **LC:PFC** was aversive and anxiogenic in both naive and neuropathic animals. This aversive action agrees with previous studies of the LC (*McCall et al., 2015*) and may reflect high tonic levels of discharge produced by the chemogenetic activation approach. Such aversion and anxiety is a common component of chronic pain syndromes (*Giamberardino and Jensen, 2015*), which likely contributes to the hypervigilance and catastrophizing. An increase in noradrenaline in the PFC has been shown in neuropathic pain models (*Suto et al., 2014*) and has been associated with an impairment of attention. Similarly, lesion studies of the **LC:PFC** have shown a reduction in pain associated anxiety and negative affect without any change in nociception (*Bravo et al., 2016*) which supports our finding that activation of **LC:PFC** produces negative affect and identifies this projection from the LC as a key player in the negative cognitive consequences of chronic pain.

These two opposing actions of the **LC:SC** and **LC:PFC** neurons to either suppress nociceptive input or to produce negative affective responses reflects the fact that they are distinct functional modules. This mirrors their anatomical segregation (*Li et al., 2016*) and implies that the modules could be

capable of independent action, with recruitment in different behavioural contexts. We speculate that it may be the balance between recruitment of the *'restorative'* LC:SC limb versus the *'alarmist'* LC:PFC projection that determines whether pain following an injury resolves or persists into chronic pain (see also [*Arora et al., 2016*; *Brightwell and Taylor, 2009*; *Taylor and Westlund, 2017*]). Further the recruitment of either module may actually reciprocally inhibit the other given the *a*2-adrenoceptor-mediated feedback inhibition within the locus coeruleus (*Aghajanian et al., 1977*). This would also require each module to have different afferent drives that were independently regulated. At present, there is little evidence for such an organisation but this is an emerging territory for which the necessary neurobiological toolboxes are only now being developed (*Schwarz et al., 2015*). Their TRIO approach using CAV2 in combination with rabies vectors suggested that there was a distinction in the afferent innervation of the *'medullary projecting'* LC neurons that we think likely represents collaterals from the LC:SC. This provides preliminary evidence that such an input – output specialisation may be present within the LC.

Previous studies have employed subtractive methods to study the role of the LC in pain regulation in animal models. These methods using toxins, knock outs or lesions to inhibit the noradrenergic system typically show subtle and sometimes contradictory effects on nociception (*Martin et al., 1999*; *Jasmin et al., 2003*, *2002*; *Tsuruoka and Willis, 1996*; *Howorth et al., 2009b*). This may be in part ascribed to the long-term adaptation to the loss of a population of neurons. However, our findings of antithetical LC actions reported herein (and *Hickey et al., 2014*) provide an alternative explanation; namely that the LC is not a single functional entity with respect to pain regulation. Rather there are counterbalancing actions of individual modules and consequently the net effect of subtractive interventions will be difficult to predict/interpret. This consideration also applies to attempts to obtain therapeutic benefit from systemic dosing of noradrenaline re-uptake inhibitors in patients – while some benefit, a majority either obtain no benefit or suffer side effects that are often related to anxiety, changes in affect and sleep disturbances that are likely due to actions on ascending LC projections. Our findings indicate that targeting just the spinal projection could separate analgesia from many of the side effects.

In demonstrating this functional and anatomical dichotomy within the LC modules, we have provided behavioural evidence that indicates that the LC has levels of sub-specialisation according to modalities. This builds upon prior work indicating anatomical and cellular specialisation within the LC (*Loughlin et al., 1986*; *Kebschull et al., 2016*; *Chandler et al., 2014*; *Howorth et al., 2009a*; *Li et al., 2016*). The LC has been implicated in diverse neuronal behaviours such as learning and memory (*Takeuchi et al., 2016*; *Martins and Froemke, 2015*), strategy (*Tervo et al., 2014*), arousal (*Carter et al., 2010*) and anxiety (*McCall et al., 2015*). For each example, the role of the LC in that behaviour has been considered as acting on a specific CNS substrate (i.e. hippocampus, auditory cortex, prefrontal cortex and amygdala) by inference somewhat in isolation. We speculate that the organisation of the LC outputs into modules allows these influences in different behavioural contexts to be independently selected and regulated. Given this organisational structure, it may be that that the *Locus Coeruleus* would be more appropriately named *Loci Coeruleus* reflecting the fact that it appears to be a collection of noradrenergic modules each with distinctive paths and roles in regulating brain function.

# Materials and methods

**Key resource table**

| Reagent or Resource | Source | Identifier |
|---|---|---|
| Antibodies | | |
| Chicken polyclonal anti-EGFP | Abcam | #AB13970 |
| Mouse monoclonal anti Dopamine-β-hydroxylase (DBH) | Chemicon | #MAB308 RRID:AB_2245740 |
| Rabbit monoclonal anti-HA tag | Cell Signalling | #C29F4 RRID:AB_10693385 |
| Chemicals, Peptides and Recombinant Proteins | | |
| (Z)—4-Hydroxytamoxifen | Sigma | H7904 |

*Continued on next page*

*Continued*

| Reagent or Resource | Source | Identifier |
|---|---|---|
| urethane (ethyl carbamate) | Sigma | #U2500 |
| PSEM308/89S | Initially: Peter Lee from Sternson group Subsequently: Custom synthesis by Apex Scientific Inc | N/A |
| Yohimbine hydrochloride | Tocris | # 1127 |
| Reboxetine mesylate | Tocris | # 1982 |
| Atipamezole hydrochloride | Pfizer, Antisedan | |
| Recombinant DNA | | |
| rAAV-syn::FLEX- rev::PSAML141F, Y115F:5HT3HC- IRES-GFP | Addgene | 32477 |
| pCAVΔE3Sce | Kremer lab | N/A |
| pCMV-EGFP-2A-PSAM HA | GeneArt AG | N/A |
| Cell lines | | |
| Lenti-X293T cells | Clonetech | # 632180, identity confirmed by manufacturer and certified as mycoplasma free |
| BJ5183 cells | Agilent Technologies | # 50-125-019, # 632180, identity confirmed by manufacturer and certified as mycoplasma free |
| DK-SceI | Kremer lab | Mycoplasma free. |
| Software and Algorithms | | |
| Ethovision ET | Noldus | N/A |
| Photoshop CS5 | Adobe | N/A |
| Illustrator CS5 | Adobe | N/A |
| Prism7 | GraphPad | N/A |
| Excel | Microsoft | N/A |
| A plasmid Editor (ApE v2.0.51) | M. Wayne Davis | http://biologylabs.utah.edu/jorgensen/wayned/ape/ |
| Spike2 v7 | Cambridge Electronic Design | N/A |
| Fiji | ImageJ | https://fiji.sc |

## Experimental model and subject details

All procedures conformed to the UK Animals (Scientific Procedures) Act 1986 and were performed under Home Office project licence (3003362) and were approved by the University of Bristol local Ethical Review Panel. Animals were housed, with an enriched environment (cardboard tubes, wood block chews, string, and varied bedding material), under a reversed 12 hr light/dark cycle at (23°C), with ad libitum access to food and water. All behavioural experiments were performed under red light conditions during the active dark phase of the cycle. Male Wistar rats from the University of Bristol colony were used for slice electrophysiology with vector injection at postnatal day 21 and subsequent use in experiments 10–14 days later. For behavioural assays and in vivo electrophysiology, male Wistar rats (120–200 g, Charles River), randomly assigned to experimental groups, and injected with the appropriate vector. The observer was blind to both the experimental groups and injected solution for all behavioural experiments. All surgical procedures were carried out under aseptic conditions and animals were maintained at 37°C on a homeothermic pad. All animals in nerve injury experiments were single housed. Brainstem tissue from all vector injected animals was histologically analysed post hoc to confirm transduction of LC neurons (EGFP expression). Out of 85 animals, 4 were excluded from the analysis because of low transduction (<10 EGFP labelled neurons) (See *Supplemental file 1* for details of animal group allocations).

## Method details
### Generation of LV$^{PRS-EGFP-2A-PSAM}$

The chemogenetic excitatory ionophore PSAM$^{L141F,Y115F:5HT3HC}$ (PSAM) (*Magnus et al., 2011*) was obtained in inverted orientation from Addgene (#32477). This sequence information was used to (1)

generate a CMV promotor driven expression plasmid by excising the PSAM-IRES-EGFP cassette with BsrGI/NheI and subsequent ligation into the pre-cut expression plasmid (BsrGI/NheI) which caused the flip in sense orientation and (2) to design (using ApE) a HA tagged (8 C-terminal amino acids) version of PSAM in the EGFP-2A-PSAM$_{HA}$ expression cassette with the objective of producing stronger EGFP fluorescence whilst PSAM receptor function is no different to the original cassette (*Figure 1—figure supplement 1*). The CMV-EGFP-2A-PSAM$_{HA}$ cassette was custom synthesised in a plasmid (pCMV-EGFP-2A-PSAM$_{HA}$ ) by GeneArt AG.

The LV was produced as previously detailed (*Coleman et al., 2003*; *Hewinson et al., 2013*). In short, the LV is based on a HIV-1 vector pseudotyped with vesicular stomatitis virus glycoprotein. The transgene product EGFP-2A-PSAM$_{HA}$ was excised with BamHI and BstBI and ligated into a pre-cut lentiviral (pTYF) backbone vector harbouring the noradrenergic-specific PRSx8 (PRS) promotor (*Hwang et al., 2001*) to generate pTYF-PRS-EGFP-2A-PSAM$_{HA}$ -WPRE (Sequence available at data. bris (DOI: 10.5523/bris.1hy0z83o0vwp62tfyboixj9szl) and plasmid from Addgene). The LVs were produced by co-transfection of Lenti-X293T cells (Clontech) with the shuttle vector pTYF-PRS-EGFP- 2A-PSAMHA-WPRE or pTYF-PRS-EGFP-WPRE, along with a packaging vector, pNHP, and the envelope plasmid pHEF-VSVG. The virus containing supernatant was collected on 3 consecutive days post transfection, pooled and concentrated by ultracentrifugation. The titre, measured in transducing units (TU/ml), was assayed by establishing the infection rate of an LV produced in parallel and harbouring the expression cassette for placental alkaline phosphatase (*Hewinson et al., 2013*). The LV was used for microinjection at a titre of $7.5 \times 10^{10}$ TU/ml for LV-PRS-EGFP-2A-PSAM$_{HA}$ and at $3.7 \times 10^{10}$ TU/ml for LV-PRS-EGFP.

## Generation of CAV2$^{PRS-EGFP-2A-PSAM}$

The EGFP-2A-PSAM$_{HA}$ cassette was excised from the custom synthesised plasmid (pCMV-EGFP-2A-PSAM$_{HA}$ ) with ApaI and HpaI. The CAV2 shuttle vector pTCAV2-PRS-ChR2-mCherry (previously described [*Li et al., 2016*]) was cut with ApaI/HpaI to remove ChR2-mCherry and subsequently ligated to generate pTCAV2-PRS-EGFP-2A-PSAM$_{HA}$. The resulting plasmids were then purified and digested with BamHI/NotI for homologous recombination into the SwaI linearized CAV2 genome (pCAVΔE3Sce) in BJ5183 cells (Agilent Technologies) following the manufacturer's protocol. CAV2 vectors were produced using previously described methods (*Ibanes and Kremer, 2013*). Briefly, DK-SceI dog kidney cells constitutively expressing the fusion protein of the endonuclease SceI and the estrogen receptor (SceI-ER) were grown in complete DMEM (with 10% fetal bovine serum, non-essential amino acids, L-glutamine, penicillin and streptamycin and G418) on 10 cm dishes to 80% confluency. Subsequently, cells were treated with 1 mM OH-tamoxifen for 4 hr to induce translocation of SceI-ER into the nucleus. The tamoxifen pre-treated DK-SceI cells ($8 \times 10^6$) were seeded in 10 cm dishes containing 1 mM OH-tamoxifen and immediately transfected with 15 µg of the super-coiled pCAV genome using 30 µl turbofect. The transfection medium was replaced by complete DMEM the following morning. Cells were harvested and virus released by freeze-thaw cycles to prepare initial virus containing stocks for vector amplification in DK-SceI cells (*Ibanes and Kremer, 2013*). Vector stocks were titrated in functional transducing units (TU) per ml as previously described (*Li et al., 2016*; *Ibanes and Kremer, 2013*; *Kremer et al., 2000*) and diluted to $0.6 \times 10^{12}$ TU before injection.

## Stereotaxic injections

The procedures for viral vector injections into the LC, the PFC or the SC have been described in detail previously (*Hickey et al., 2014*; *Howorth et al., 2009a*; *Li et al., 2016*). In brief, rats were anesthetized by intraperitoneal injection of ketamine (5 mg/100 g, Vetalar; Pharmacia, Sweden) and medetomidine (30 µg/100 g, Domitor; Pfizer, UK) until loss of paw withdrawal reflex. The rat was placed in a stereotaxic frame. Burr holes and microinjections were made using a Drill and Injection Robot with Wireless Capillary Nanoinjector (NEUROSTAR, Germany).

Locus coeruleus targeting in rat pups (p19-21): A dorso-ventral injection track was started at 1 mm lateral and 1 mm posterior to lambda with 10˚ rostral angulation. Four injections of 0.25 µl of vector were made at four locations beneath cerebellar surface (−4.6 mm, −4.9 mm, −5.2 mm and −5.5 mm).

Locus coeruleus targeting in adult rats (120–200 g): A dorso-ventral injection track was started at 1.2 mm lateral and −2.1 mm posterior to lambda with 10° rostral angulation. Three injections of 0.3 µl were made at −5.3 mm, −5.5 mm and 5.8 mm depth. In rats intended for in vivo LC recordings a screw was inserted into the burr hole to simplify later CNS access.

Prefrontal cortex: Four burr holes were drilled bilaterally, 0.7 mm lateral to the midline, at +2 mm, +1.2 mm, +0.8 mm and −0.2 mm from bregma. At each site 0.2 µl of vector was injected at 1.0 mm and 1.5 mm depth to the brain surface. At + 2 mm further injections were made at −5 and −5.5 mm depth from the brain surface

Spinal cord: The vertebral column was exposed and clamped at vertebral segment L3. A laminectomy was performed at T12-L1 and 0.4 µl CAV2 vector injected bilaterally into the L3-L4 segment of the spinal cord (SC) at six injection sites in total. These injections were located 0.5 mm lateral from the midline and 0.5 mm deep and spread in pairs over the segmental extent. *For unilateral injections (pain studies)* six injections of (0.4 µl) were made unilaterally into the right lumbar dorsal horn.

## Electrophysiology

### Patch clamp recordings from acute pontine slices

Pontine slices were prepared from juvenile (age p28-p32) Wistar rats 7–14 days after lentiviral transduction as previously described (*Hickey et al., 2014*). Rats were deeply anesthetized with isoflurane 5%, decapitated and the brain quickly removed and immediately chilled in ice-cold cutting solution (similar to the recording solution but NaCl was reduced from 126 mM to 85 mM and substituted with sucrose 58.4 mM) and the brainstem blocked. Horizontal slices (300–350 µm thick) of the pons were cut from dorsal to ventral using a vibratome (Linearslicer Pro 7; DSK) in cold (4°C) cutting solution. After cutting, slices were kept at room temperature in carbogenated recording solution (NaCl 126 mM, KCl 2.5 mM, NaHCO$_3$ 26 mM, NaH$_2$PO$_4$ 1.25 mM, MgCl$_2$ 2 mM, CaCl$_2$ 2 mM, and D-glucose 10 mM saturated with 95%O$_2$/5%CO$_2$, pH7.3, osmolality 290 mOsm/L) for at least 1 hr to recover before recording. Pontine slices were transferred into the recording chamber of an upright fluorescence microscope (DMLFSA; Leica Microsystems), superfused with artificial CSF at a rate of 4–8 ml/min heated to 35°C. Borosilicate glass patch pipettes (Harvard Apparatus GC120F-10, resistances of 4–6 MΩ) were filled with internal solution (K-gluconate 130 mM, KCl 10 mM, Na-HEPES 10 mM, MgATP 4 mM, EGTA 0.2 mM, and Na$_2$GTP 0.3 mM). Cells were identified under gradient contrast illumination and examined for transduction by epifluorescence illumination. Whole cell voltage and current clamp recordings were obtained with a Multiclamp 700 A amplifier (Axon instruments). All membrane potentials were corrected for a junction potential of 13 mV. Drugs were infused in the bath solution. Spike2 software (CED, Cambridge Electronic Design) was used to acquire and store data.

### In vivo extracellular recordings from the locus coeruleus and local application of PSEM308

Anaesthesia was induced with 1.2–2 g/kg urethane (Sigma) i.p to achieve loss of paw withdrawal reflex. Once induced, urethane anaesthesia typically remained stable for 5–7 hr. The animal was placed in a stereotaxic frame, the skull exposed and the screw above the viral injection site removed to gain access to the CNS. In vivo single units were recorded with preloaded quadruple-barrelled microelectrodes (Kation Scientific, Carbostar-4 recording/iontophoresis microelectrode, E1041 standard) connected to a drug injector (Parker, Picospritzer II). Recordings were referenced against a silver chloride electrode pellet that was inserted between muscles and skin near the recording site. Neuronal activity was recorded via a Multiclamp 700A amplifier (with #CV-7b headstage, Axon instruments) filtered at 3 kHz and digitized at 10 kHz using a 1401micro and Spike2 software (CED, Cambridge Electronic Design) was used to store data and to control the drug delivery. Locus coeruleus cells were identified by their wide (~1 ms), large amplitude action potentials, the slow spontaneous firing frequency (0.5–5 Hz) and a characteristic burst of action potentials in response to contralateral hind-paw pinch (*Hickey et al., 2014*; *Cedarbaum and Aghajanian, 1976*). Action potential discharge frequency was measured in 5 s bins and compared before, during, and after drug application. The average depth of electrode placement for LC recordings was 5.7 ± 0.14 mm from the cerebellar surface.

## In vivo single unit recordings from the dorsal horn

Animals were terminally anaesthetised with urethane (1.2–2 g/kg i.p, Sigma) The spinal cord was exposed by laminectomy over the L1-L4 spinal segment for recordings (*Funai et al., 2014*; *Furue et al., 1999*). The animal was placed in a stereotaxic frame and the spinal cord stabilised with two spinal clamps at T1 and L5 and a bath was formed by skin elevation. Warmed (36°C), Krebs solution was used to continually superfuse the spinal cord (NaCl 117 mM, KCl 3.6 mM, NaHCO$_3$ 25 mM, NaH$_2$ PO$_4$ 1.2 mM, MgCl$_2$ 1.2 mM, CaCl$_2$ 2.5 mM and D-glucose 11 mM saturated with 95%O$_2$ /5% CO$_2$ , pH7.3, osmolality 290 mOsm/L). The dura mater was removed under binocular vision (Leica MZ6). A reference electrode (World Precision Instruments, Ag/AgCl Electrode pellets, EP2) was inserted into the muscle layer near the recording area and spinal units were recorded with a stainless-steel microelectrode (FHC, UESSEGSEFNNM, 8–8.8 MΩ) that was positioned ~500 μm from the midline. Neuronal activity was amplified and low-pass filtered at 3 kHz using a Multiclamp 700A amplifier (Axon instruments with #CV-7b headstage). The data were digitized at 10 kHz (CED Power1401) and stored using Spike2 software (CED, Cambridge Electronic Design). Pinch was applied from custom calibrated forceps with tip pads of area 3.5 mm$^2$. These forceps were used to ensure that the pinch stimulus was standardised between applications and across animals. The forceps were equipped with strain gauges and a pinch pressure of 70 g mm$^{-2}$ (*Drake et al., 2014*) was used throughout all experiments to compare pinch-evoked neuronal activity (NB this pressure was above the threshold for pain when applied to the experimenter's index finger).

Wide dynamic range (WDR) neurons were identified in the deep dorsal horn by graded responses to both non-noxious touch and noxious pinch. Their discharge frequency was measured in response to light touch with a brush or to sustained pinch pressure over five second time bins immediately after stimulus application. Extracellular recording data from four spinal cord neurons were excluded from analysis because the recording was interrupted before completion of the experimental protocol.

## Tibial nerve transection model

Rats were anesthetized by intraperitoneal injection of ketamine (5 mg/100 g, Vetalar; Pharmacia, Sweden) and medetomidine (30 μg/100 g, Domitor; Pfizer) until loss of paw withdrawal reflex. A 1.5–2 cm long skin incision was made in the right hind leg in line with the femur (*Pertin et al., 2012*). The fascial plane between the gluteus superficialis and the biceps femoris was located and blunt dissected to expose the three branches of the sciatic nerve (i.e. common peroneal, sural and tibial nerves). The tibial nerve was carefully exposed. Subsequently, two tight ligations (5–0 silk) were made approximately 2 mm apart and the neve segment in between transected and removed. The nerve and muscle were placed back into the correct position and the skin sutured. For sham surgery, the same procedure without ligation and transection of the tibial nerve was performed.

## Intrathecal injection after nerve injury

The injection procedure was similar to a previous report (*De la Calle and Paíno, 2002*) Rats, 6 weeks post-TNT (400–500 g), were briefly induced with isoflurane in medical oxygen before transferring into a face mask with 1.5–2% isoflurane. The rat was placed in prone position on a 5 cm tall plastic board so that the hind legs hang over the board to arch the lower back and to open the intervertebral space between L4-L5 spinal segments. Subsequently, a 25G needle was inserted transcutaneously to the vertebral canal (typically producing a tail twitch) and 10–20 μl of vehicle or yohimbine (3 μg/μl in sterile saline with 20% DMSO) injected. PSEM308 was injected i.p immediately after recovery from anaesthesia and sensory testing began 30 min thereafter.

As previously described (*Hughes et al., 2013*), we observed that the mechanical withdrawal thresholds fell in the contralateral paw after intrathecal injection of yohimbine in nerve injured animals. This phenomenon was used as an internal control for successful intrathecal targeting because it was more reliable than a tail flick response (see *Figure 5—figure supplement 1*).

## Behaviour

Rats underwent two 30 min habituation sessions (unless otherwise stated) on consecutive days before testing. Animals were placed into Plexiglas chambers 20 min before sensory assessment

began on the test day. All experiments, apart from conditioned place preference (CPP), were conducted in a cross-over design comparing saline and PSEM308 (10 mg/kg, unless otherwise specified). The experimenter was always blind to the animal group and to the administered solution.

## Hargreaves' test

Thermal withdrawal latencies were measured for the hind paw using the method described by *Hargreaves et al. (1988)*. Animals were placed into a Plexiglas chamber on top of a glass plate. The radiant heat source (Ugo Basile Plantar test) was focused onto the plantar surface of the hind paw and the time to withdrawal recorded from 40 min before and up to 60 min after intraperitoneal injection of the selective agonist PSEM308 or saline. Each withdrawal value was the mean of two tests and tests were repeated every 15 min. A 24 s cut-off value was used to avoid tissue damage. The heat source intensity was adjusted so that control rats responded with a latency of 7–8 s.

## Spontaneous foot lifts

Animals were placed into a Plexiglas chamber on a metal grid and a webcam (Logitech HD Pro Webcam C920) positioned underneath. The number of spontaneous footlifts and flinches was recorded over a 5-min period 20 min after i.p injection of PSEM308/saline.

## von Frey test

To assess punctate mechanical sensitivity animals were placed into a Plexiglas chamber on a metal grid and von Frey filaments (0.4 g-26g) were applied to the lateral aspect of the plantar surface of the hind paw. The 50% paw withdrawal threshold was determined using the Massey-Dixon up-down method (*Chaplan et al., 1994*). This procedure was repeated every 30 min starting 1 hr prior and up to 4 hr after i.p injection of PSEM308 (1, 5 and 10 mg/kg) or saline.

## Acetone test

To assess cold sensitivity animals were placed into a Plexiglas chamber on a metal grid and an acetone drop on top of a 1 ml syringe was presented to the hind paw from underneath five times, 2 min apart. Data are expressed as the percentage of withdrawals. Animals were tested prior to and 45 min after i.p injection of PSEM308 or saline.

## Incapacitance test

The distribution of weight between the hind-limbs was measured with an incapacitance tester (Linton Instrumentation). Animals were habituated to the Plexiglas chamber with their hind paws on two force transducers in four sessions (each lasting 3 min) on 2 consecutive days. Measures were taken 45 min after injection of PSEM308 or saline. The percentage of weight borne on the injured leg was averaged over these four measures.

## Conditioned place preference/aversion

All rats were habituated to an unbiased three-compartment box, which contained a central neutral chamber with different association chambers either side. Note, two designs of box were employed the first for naive animals had a central chamber between the two conditioning chambers however in the course of these experiments two animals learnt how to escape the arena (and were excluded). Therefore, we modified this arrangement with higher walls and an offset connecting corridor between the two otherwise similar conditioning chambers for the nerve injured animals which adequately contained the animals and reduced the amount of time spent in the neutral chamber.

The conditioning compartments have visual and tactile cues e.g. 'bars' or perforated 'holes' as flooring and differently patterned (but equal luminosity) wallpapers to maximise differentiation between the compartments. On the first day of the experiment animals were allowed to roam freely through all three compartments and a Basler camera (acA1300-60 gm) with a varifocal lens (Computar H3Z4512CS-IR) connected to EthovisionXT (Noldus) was used to record the time rats spent in either of the outer compartments as a baseline value. Subsequently, rats underwent four conditioning sessions in which the outside compartments were paired with PSEM308 injection or saline injection. Two pairing sessions per day at least 4 hr apart on two consecutive days were conducted in a counterbalanced design. For each conditioning session the animals were placed into the locked

conditioning compartment 5 min after the i.p injection and left for 30 min. On the test day, the doors between the compartments were opened and the time spent in the conditioning compartments measured once more. It is expected that the choice of the animal's preferred environment in the post-test is influenced according to the valence of the chemogenetic stimulation. Noldus EthoVision XT was used to analyse recorded videos and to quantify the time animals spent in the paired chambers.

## Open-field test
On the first day of testing animals were randomly assigned into two groups and received PSEM308 or saline 20–25 min before placing the subjects into the open-field arena (56.5 × 56.5 cm with an inner zone of 30 × 30 cm) for 6 min. Exploration behaviour (distance travelled, location in central/ perimeter areas and immobility (velocity < 1.75 cm/s)) was tracked for 5 min starting 60 s after entering the arena using EthoVisionXT. One week later the open-field experiment was repeated using the other solution.

## Rotarod test
Nerve injured animals were trained to balance on a rotating rod over 3 days consisting of 2–3 5-min-long training sessions per day. Subsequently, animals were assigned randomly into two groups and received PSEM308 or saline 20–25 min before placing the subjects onto the rod. The latency to fall was measured at a constant speed of 9 rotations/min. One day later, the rotarod experiment was repeated using the other drug solution.

## Histology
### Tissue collection and processing
Rats were killed with an overdose of pentobarbital (Euthatal, 20 mg/100 g, i.p; Merial Animal Health) and perfused transcardially with 0.9% NaCl (1 ml/g) followed by 4% formaldehyde (Sigma) in phosphate buffer (PB; pH 7.4, 1 ml/g). The brain and spinal cord were removed and post-fixed overnight before cryoprotection in 30% sucrose in phosphate buffer. Coronal tissue sections were cut at 40 μm intervals using a freezing microtome and left free floating for fluorescence immunohistochemistry.

### Immunofluorescence
Tissue sections were blocked for 45 min in phosphate buffer containing 0.3% Triton X-100 (Sigma) and 5% normal goat serum (Sigma). Incubated on a shaking platform with primary antibodies to detect EGFP (1:3000), DBH (1:2000), or HA tag (1:500-1:1000) for 14–18 hr at room temperature. After washing, sections were then incubated for 3 hr with appropriate Alexa Fluor secondary antibodies. A Leica DMI6000 inverted epifluorescence microscope equipped with Leica DFC365FX monochrome digital camera and Leica LAS-X acquisition software was used for widefield microscopy. For confocal and tile scan confocal imaging, a Leica SP5-II confocal laser-scanning microscope with multi-position scanning stage (Märzhäuser, Germany) was utilized. Images were exported as TIF files before being processed, analysed and prepared for presentation using Fiji- ImageJ software.

Stereological cell counts (*Miller et al., 2014*) were made by stepping through a confocal z-stack with 1 μm spacing and EGFP positive cells were only counted if the top of the soma was unambiguously located within the imaged dissector volume of the section (12 μm). The average section thickness after histological processing was $18.0 \pm 0.7$ μm (30 sections from three animals) and dual guard zones of 2 μM were applied. Therefore, the height sampling factor was 0.66 (hsf = dissector height/ average section thickness). The section sampling factor was 0.25 (ssf = 1/ number of serial sections). The correction was made using the following formula.

$$N = C * \frac{1}{volume\ fraction}$$

where N is the estimated number of cells and C the sum of all counted cells. The volume fraction is the product of ssf, hsf and asf (area sampling fraction). Because all EGFP-positive neurons in the LC area were counted the area sampling fraction was omitted from the volume fraction.

The percentage of EGFP-labelled noradrenergic (DBH) axons in the spinal cord and in cortical regions were quantified in 82 μm x 82 μm x 10 μm z-stacks with 0.5 μm z-spacing. Three z-stacks from different brain sections per area of interest were averaged.

## Quantification and statistical analysis

All statistical analyses were conducted using GraphPad Prism 7. All data are presented as mean ± SEM (unless otherwise specified). Sample size was estimated from previous experience and was similar to those generally employed in the field (*McCall et al., 2015*; *King et al., 2009*; *Hughes et al., 2015*). T-tests, one- way or two-way ANOVA were used to compare groups as appropriate. Bonferroni's post-test was used for all comparisons between multiple groups. Differences were considered significant at $p < 0.05$. The number of replications (n) equals the number of cells for electrophysiological experiments (the number of individual rats or preparations is stated in the relevant section of the text), (N) number of animals tested in behavioural assays (each experiment was repeated with at least two independent batches of animals for each condition) and the number of LCs that were analysed in histological experiments.

## Contact for reagent and resource sharing

Further information and requests may be directed to, and will be fulfilled by, Dr. Anthony E. Pickering. The plasmid sequence LV: PRS-EGFP-2a-PSAM HA is available online from data.bris at: DOI: 10.5523/bris.1hy0z83o0vwp62tfyboixj9szl and the plasmid will be deposited with Addgene. CAV2 plasmids and vector will be available from the vector core facility in Montpellier (Eric J Kremer).

## Acknowledgements

This work was supported by: Wellcome Trust Senior Clinical Research Fellowship (gr088373) – AEP University of Bristol Postgraduate Research Scholarship – SH. EMBO short-term fellowship programme to facilitate researcher exchange (SH) between Kremer and Pickering lab The authors thank: *The Sternson lab at Janelia Research Campus* for kind gifts of PSEM89s/PSEM308 for initial trials, plasmids containing PSAM sequences and for technical advice Hidemasa Furue and Robert Drake for advice on electrophysiological experiments and data presentation Sandy Ibanes for help with vector production

## Additional information

### Funding

| Funder | Grant reference number | Author |
| --- | --- | --- |
| Wellcome | gr088373 | Anthony E Pickering |
| University of Bristol | | Stefan Hirschberg<br>Andrew Randall<br>Anthony E Pickering |
| European Molecular Biology Organization | | Stefan Hirschberg<br>Eric J Kremer<br>Anthony E Pickering |

The funders had no role in study design, data collection and interpretation, or the decision to submit the work for publication.

### Author contributions

Stefan Hirschberg, Conceptualization, Data curation, Formal analysis, Investigation, Visualization, Methodology, Writing—original draft, Writing—review and editing; Yong Li, Resources, Supervision, Investigation, Methodology, Writing—review and editing; Andrew Randall, Conceptualization, Formal analysis, Supervision, Methodology, Writing—review and editing; Eric J Kremer, Conceptualization, Resources, Supervision, Funding acquisition, Methodology, Writing—review and editing; Anthony E Pickering, Conceptualization, Resources, Data curation, Formal analysis, Supervision,

Funding acquisition, Validation, Investigation, Visualization, Methodology, Writing—original draft, Project administration, Writing—review and editing

### Author ORCIDs

Anthony E Pickering http://orcid.org/0000-0003-0345-0456

### Ethics

Animal experimentation: All procedures conformed to the UK Animals (Scientific Procedures) Act 1986, were performed under Home Office project licence (3003362) and were approved by the University of Bristol Animal Welfare and Ethical Review Board.

### Decision letter and Author response

Decision letter https://doi.org/10.7554/eLife.29808.026
Author response https://doi.org/10.7554/eLife.29808.027

## Additional files

### Supplementary files

• Supplementary file 1. List of animals used in electrophysiological and behavioural studies.
DOI: https://doi.org/10.7554/eLife.29808.024

• Transparent reporting form
DOI: https://doi.org/10.7554/eLife.29808.025

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
