## [Decision Letter]

Thank you for submitting your article "Engagement of spinal- vs prefrontal-projecting locus coeruleus neurons parses analgesia from aversion & anxiety in rats" for consideration by *eLife*. Your article has been reviewed by three peer reviewers, one of whom is a member of our Board of Reviewing Editors and the evaluation has been overseen by Gary Westbrook as the Senior Editor. The reviewers have opted to remain anonymous. The reviewers have discussed the reviews with one another and the Reviewing Editor has drafted this decision to help you prepare a revised submission.

Summary:

This manuscript is a comprehensive and interesting manuscript that examines the functional organization of different regions of the locus coeruleus. The authors assessed the consequences of selective chemogenetic activation of the nor-adrenergic LC projections to the spinal cord or to the prefrontal cortex on modulation of pain and in eliciting either positive or negative affective states. The viral approach to localizing virus to descending vs PFC projections is impressive and convincing. The authors evaluated the animals in a neuropathic pain model produced by tibial nerve transection, which results in a mechanical and cold hypersensitivity.

Essential revisions:

In the course of your revision, we ask that additional experimental information be provided.

1) There is a discrepancy, albeit an interesting one, between the effect of LC stimulation in the CPA test, indicative of an effect on ongoing pain, and the lack of effect of LC stimulation on spontaneous activity of dorsal horn neurons. In a revised submission we ask that you examine the effect of LC stimulation on spontaneous and evoked dorsal horn activity in the nerve injury model, which will significantly supplement the findings in the absence of injury.

2) In addition, in order to ensure that this is a noradrenergic action, we ask the behavioral effects of atipamezole be tested in the injury condition.

3) Lastly, you note that there was no difference in total distance traveled in the open field test, but that doesn't really measure subtle changes in motor activity. Subtle motor dysfunction could incorrectly leave an impression of anti-nociception. A more appropriate assay for subtle changes in motor function would entail the rotorod. This experiment is essential in any pain tests where reflex withdrawal endpoints are used, and likely also when counting spontaneous lifts.

4) Although the reporting requirements of ARRIVE are largely addressed, there is no explanation for widely different sample sizes in simple, two-group experiments. Whether and how many animals were used in multiple studies is not mentioned. The reporting would be significantly improved by inclusion of CONSORT diagrams for animals in the study overall and within individual experiments in Supplementary Data. Please explicitly state that experiment-wide error, even with outcomes which would be expected to co-vary, was not considered in this study. Although you need not present this, one could calculate the likelihood that exact replication of the conditions by yourself in your own lab of all these experiments would have a <10% chance of finding all the results which are currently statistically significant to still be significant.

---

## [Author Response]

Essential revisions1) There is a discrepancy, albeit an interesting one, between the effect of LC stimulation in the CPA test, indicative of an effect on ongoing pain, and the lack of effect of LC stimulation on spontaneous activity of dorsal horn neurons.

We don’t see this as being a discrepancy. In the naïve animal (without nerve injury) there is no noxious stimulation under resting conditions so the ongoing activity in the WDR neurons in the anaesthetised state does not represent input from nociceptors. Therefore, there is no ongoing activity from nociceptors for activation of the LC to block (only the evoked responses to noxious stimuli). In the behavioural tests in naïve animals, activation of the LC neurons projecting to the spinal cord (LC^:SC^) shows no effect in the conditioned place aversion assay again because there is no spontaneous pain to block – so no positive affective consequence. We believe that these findings therefore are entirely consistent.

In a revised submission we ask that you examine the effect of LC stimulation on spontaneous and evoked dorsal horn activity in the nerve injury model, which will significantly supplement the findings in the absence of injury.

Notwithstanding the points made above we have undertaken this additional investigation and have added the cell recording data and an additional figure to the manuscript (Figure 6). We show in nerve injured animals that WDR neurons in the dorsal horn (n=21) respond to brush, pinch, cold and von Frey hairs with increases in activity (as would be expected given the allodynic sensitisation). These evoked responses are all reversibly diminished by activation of the LC^:SC^ projection. This attenuation is blocked by atipamezole. This data is now included in the results (subsection “Engagement of LC^:SC^ module attenuates neuropathic sensitisation via α2-adrenoceptors”) and significantly supplements our findings. Note we did not find any effect on spontaneous discharge suggesting that the LC^:SC^ is acting to suppress afferent inputs to the WDR but is not causing direct inhibition of the WDR neurons themselves.

2) In addition, in order to ensure that this is a noradrenergic action, we ask the behavioral effects of atipamezole be tested in the injury condition.

We have now added tests of the effect of atipamezole (50µg) on the behavioural responses. We had already tested the α2-antagonist Yohimbine in this paradigm and have now shown that Atipamezole, a more selective, but shorter acting, α2–antagonist has an identical effect on behaviour in blocking the effects of LC^:SC^ activation on evoked responses in nerve injured animals (Figure 8). This is in agreement with the new data that we have also provided from the cell recordings in TNT animals where the effect of LC^:SC^ activation is blocked by atipamezole (Figure 6). This provides robust evidence that these effects are mediated by α2 adrenoceptors and is consistent it being a noradrenergic action.

3) Lastly, you note that there was no difference in total distance traveled in the open field test, but that doesn't really measure subtle changes in motor activity. Subtle motor dysfunction could incorrectly leave an impression of anti-nociception. A more appropriate assay for subtle changes in motor function would entail the rotorod. This experiment is essential in any pain tests where reflex withdrawal endpoints are used, and likely also when counting spontaneous lifts.

We have now undertaken rotarod testing. The tibial nerve injured animals have an impaired ability to perform the rotarod test – as we would expect. The activation of the LC^:SC^ neurons has no significant effect on their ability to perform the test (Figure 5). We do not therefore find any evidence for a motor impairment that could account for the change in behaviour with LC^:SC^ activation, strengthening the case that it is indeed an analgesic action.

4) Although the reporting requirements of ARRIVE are largely addressed, there is no explanation for widely different sample sizes in simple, two-group experiments.

Unfortunately, these are not simple experiments given the need for vector injections +/- nerve injury with the repeated administration of drugs and behavioural tests over a period of 4–7 weeks. The main reason for the differences in group sizes is that some animals failed to complete a part of the planned protocol or were found on post-hoc histology to have not been successfully transduced. Also, the power to resolve differences varied across the scales of investigation from single cells to whole animal behaviour which meant that different group sizes were required.

Whether and how many animals were used in multiple studies is not mentioned. The reporting would be significantly improved by inclusion of CONSORT diagrams for animals in the study overall and within individual experiments in Supplementary Data.

CONSORT diagrams would not work especially well for our study design but we have included this data in a set of animal usage tables that transparently indicates how many animals entered each study protocol and which ones were excluded with reasons (N=4 in total for poor or absent transduction and N=2 from one test protocol).

Please explicitly state that experiment-wide error, even with outcomes which would be expected to co-vary, was not considered in this study. Although you need not present this, one could calculate the likelihood that exact replication of the conditions by yourself in your own lab of all these experiments would have a <10% chance of finding all the results which are currently statistically significant to still be significant.

We appreciate the reviewer’s point regarding reproducibility which is an important topic for biomedical researchers across all disciplines. However, we note that we are not testing multiple different null hypotheses in this study. In essence, we have a single null hypothesis which is that there is no difference in the action of the LC neurons projecting to the PFC and spinal cord on pain related behaviour. We then perform multiple experiments to assess whether we can reject this null hypothesis. Therefore, most of the tests are far from independent and are closely interrelated. Indeed, one could flip this train of thought around ask ‘would our findings be undermined if one or two or even three of our statistically significant results were to breach the 5% threshold and therefore be lost?’ We would argue that even in this worst case they would likely would not change our basic conclusions that the null hypothesis is to be rejected based on the multiple other lines of evidence indication orthogonal differences between the subsets of LC neurons, but it would make us slightly more cautious in our interpretation. We believe that this is the value of the combination of methods within the study rather than a weakness as the reviewer’s comment seems to imply. Furthermore, this study is a continuation of a line of experiments in our group (and those of several other labs) since 2009 in our case that have provided multiple lines of experimental evidence that have been building towards this study – we have indeed, in the course of this study, been able to replicate and extend several of our key previous findings. Note also that many of our analyses have used ANOVA and post-hoc testing which takes account of the multiple testing issue. On this basis, even though our findings reveal a remarkable influence of noradrenergic neurons on nociception and negative affect we have confidence that these results are robust and will be replicable both by ourselves, as we endeavour to take these ideas towards translation, and by others.